# Hybrid dedicated and distributed coding in PMd/M1 provides separation and interaction of bilateral arm signals

**Tanner C. Dixon**[1]*, **Christina M. Merrick**[2], **Joni D. Wallis**[1,2,3], **Richard B. Ivry**[1,2,3], **Jose M. Carmena**[1,3,4]

**1** UC Berkeley–UCSF Graduate Program in Bioengineering, University of California-Berkeley, Berkeley, California, United States of America, **2** Department of Psychology, University of California-Berkeley, Berkeley, California, United States of America, **3** Helen Wills Neuroscience Institute, University of California-Berkeley, Berkeley, California, United States of America, **4** Department of Electrical Engineering and Computer Sciences, University of California-Berkeley, Berkeley, California, United States of America

* tcd44@berkeley.edu

**Data Availability Statement:** All data were deposited on Dryad digital repository: https://doi.org/10.6078/D1FM6S.

## Abstract

Pronounced activity is observed in both hemispheres of the motor cortex during preparation and execution of unimanual movements. The organizational principles of bi-hemispheric signals and the functions they serve throughout motor planning remain unclear. Using an instructed-delay reaching task in monkeys, we identified two components in population responses spanning PMd and M1. A "dedicated" component, which segregated activity at the level of individual units, emerged in PMd during preparation. It was most prominent following movement when M1 became strongly engaged, and principally involved the contralateral hemisphere. In contrast to recent reports, these dedicated signals solely accounted for divergence of arm-specific neural subspaces. The other "distributed" component mixed signals for each arm within units, and the subspace containing it did not discriminate between arms at any stage. The statistics of the population response suggest two functional aspects of the cortical network: one that spans both hemispheres for supporting preparatory and ongoing processes, and another that is predominantly housed in the contralateral hemisphere and specifies unilateral output.

## Author summary

The motor cortex of the brain primarily controls the opposite side of the body, yet neural activity in this area is often observed during movements of either arm. To understand the functional significance of these signals we must first characterize how they are organized across the neural network. Are there patterns of activity that are unique to a single arm? Are there other patterns that reflect shared functions? Importantly, these features may change across time as motor plans are developed and executed. In this study, we analyzed the responses of individual neurons in the motor cortex and modeled their patterns of co-activity across the population to characterize the changes that distinguish left and right arm use. Across preparation and execution phases of the task, we found that signals

**Funding:** This work was funded by: National Institute of Health grant R01 NS097480 (JMC); National Institute of Mental Health R01 MH117763 (JDW); DoD National Defense Science and Engineering Graduate (NDSEG) Fellowship (TCD). The funders had no role in study design, data collection and analysis, decision to publish, or preparation of the manuscript.

**Competing interests:** The authors have declared that no competing interests exist.

became gradually more segregated. Despite many neurons modulating in association with either arm, those that were more dedicated to a single (typically contralateral) limb accounted for a disproportionately large amount of the variance. However, there were also weaker patterns of activity that did not distinguish between the two arms at any stage. These results reveal a heterogeneity in the motor cortex that highlights both independent and interactive components of reaching signals.

## Introduction

In the primate cortex, direct control of arm movement is primarily mediated by contralateral descending projections [1–3]. However, numerous studies have observed activity changes in the motor cortex during movements of the ipsilateral arm [4–11] and hand [12–14]. The functional role of this ipsilateral activity has been the subject of considerable debate, with hypotheses ranging from a role in postural support, bimanual coordination, or an extrapyramidal control signal for unimanual movements.

Neurons in the primate dorsal premotor cortex (PMd) play a critical role in motor preparation [6,8,12,15,16]. Interestingly, their response properties and degree of laterality appear to change across the course of preparation. For example, within PMd, individual units exhibit a transition from effector-independent to effector-dependent encoding between preparatory and execution phases of reaching. In contrast, units in primary motor cortex (M1) mainly become active during movement itself and show a pronounced contralateral bias [8]. This suggests a transition from abstract planning to explicit specification of motor output parameters in the signals of individual neurons. A similar transition has been shown in the activation of different cell-types within rodent motor areas [17,18]. These studies have found that neurons with intracortical projections show little lateral bias, particularly during pre-movement phases. In contrast, neurons with descending output display much stronger laterality, especially just before and during movement. This adds yet another level of granularity in the discussion of lateralized motor function. Collectively, these single-unit studies support a notion that there exist two distinct components within the motor cortex: one that is bilateral and likely involved in abstract processing, and another that is dedicated to a single side of the body for execution.

The classical perspectives outlined above have been revisited in studies that focus on population-level analysis, considering instead how computations might be reflected in the way the network coordinates activity. Low-dimensional representations of large-scale neural recordings can be used to characterize these network patterns, revealing changes in covariance structure across behavioral settings that are not evident when looking at single neurons in isolation [19]. Ostensibly, these changes reflect reorganization of the population as it engages in different computational processes. Using these methods, pre-movement activity has been shown to evolve within an "output-null" subspace towards an optimal initial population state [20–22]. This initial state is advantageously positioned for engaging the internal dynamics of the network to produce patterned activity in an "output-potent" subspace for driving movement [23–25]. There is some evidence that bilateral activity may support these preparatory and dynamic properties [26,27]. Similar to the output-null and output-potent subspaces, arm-specific subspaces have also been observed in M1 during rhythmic movements [10] and in response to joint perturbations [11]. It remains unclear precisely what organizational principles produce these arm-specific subspaces, whether signals are fully separated at the level of the population, and how such properties develop across preparation and movement.

There are two fundamental and mutually non-exclusive ways that population signals may specify the selected arm across preparation and movement. (1) Signals may consolidate within dedicated sub-populations for each arm (i.e., within hemispheres, brain areas, or cell-types). (2) Signals for each arm may be distributed across the same units yet maintain unique covariance structure that separates them along arm-specific neural dimensions. Importantly, either of these architectures provides a way for downstream targets to discriminate signals and also yields the mathematical result of divergent subspaces. The first method is necessarily true: contralateral biases have been consistently observed during movement and, to a lesser extent, during preparation as well. Such lateral biases will trivially orthogonalize arm signals. The question is whether the second method is also true. Either signals that are mixed within units become separated (arm-specific) in the population readout, or they exist within a space where the same patterns of activity are involved in computations relevant to both arms. This is a vital distinction to make, as it constrains the possible roles that bilateral activity can play at each stage of processing and may point to an important heterogeneity in the population statistics.

In the present study, we recorded large populations of single-units in PMd and M1 bilaterally while monkeys performed an instructed-delay unimanual reaching task. As activity emerged during preparation, there was a tendency for units with stronger arm preference to be more highly modulated, therefore accounting for a larger proportion of the population variance. As a result, the signals for each arm were largely segregated, primarily within contralateral PMd. During the transition to movement, M1 became more prominently involved and the signals for each arm became increasingly segregated. This unit-level segregation caused the subspaces corresponding to each arm to diverge across the trial. However, we did observe target-specific information that was not only present in these segregated signals, but also mixed within individual units, indicating incomplete separation of control signals for each side. Importantly, the subspace containing this information did not segregate signals for the two arms at any stage. Taken together, the results point to two primary components in the population response: (1) A dedicated component that develops across preparation, reaches a maximum during movement, and mirrors the lateralized anatomy of corticospinal output with its contralateral bias. (2) A distributed component that represents far less variance, particularly during movement, and provides a space in which bilateral control signals coexist and may readily interact.

## Results

### Behavior

Two macaque monkeys were trained to perform an instructed-delay reaching task in 3-D space (Fig 1A). Reaching movements were freely performed in an open area while kinematics were recorded using optical motion tracking. Visual feedback of endpoint position and task cues were provided through a virtual 3-D display. Each trial had three phases (Fig 1B). For the Rest phase, the monkey placed both hands in start targets positioned near the torso and remained still for 500 ms. For the Instruct phase, an instructional cue appeared at one of six target locations. The color of the cue specified the required hand for the forthcoming trial. The monkey was required to keep both hands in the rest positions while the cue remained visible for a variable interval (500–1500 ms). The Move phase was initiated when the start position marker for the reaching hand disappeared and the cue at the target location increased in size, which signaled the animal to reach. The monkey received a juice reward if it accurately reached the target and maintained the final position for 250 ms, while keeping the non-cued hand at its start position for the duration of the trial. 300ms representative windows from each phase were used in data analysis.

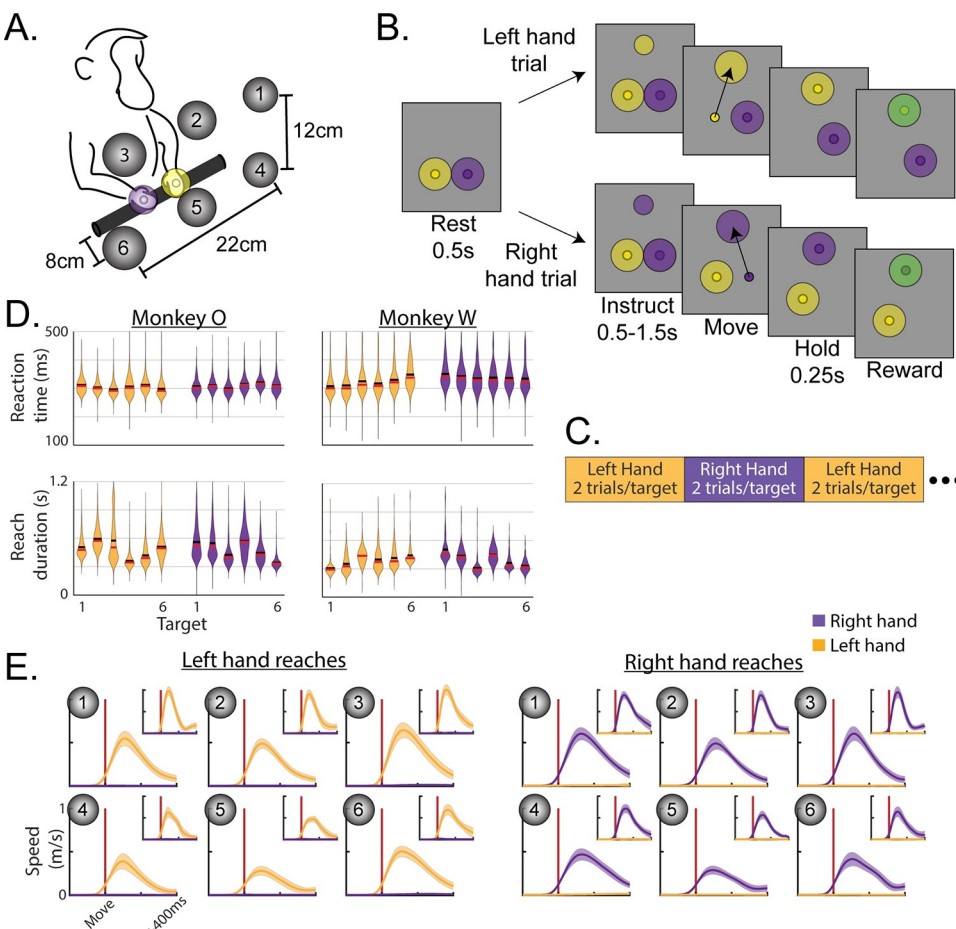

**Fig 1. Behavior.** (**A**) Monkeys reached to one of six virtual targets, indicated by grey spheres in the cartoon. During the task these would be invisible until one appeared to instruct the reach. (**B**) Trials consisted of 3 phases. Each trial was initiated by placing both hands in start targets and remaining still for 500ms (Rest phase). A small target then appeared at the location of the future reach in a color that indicated which hand to use. The monkey remained still during cue presentation for 0.5–1.5s (Instruct phase). The start target for the reaching hand then disappeared while the reach target enlarged to cue movement (Move phase). (**C**) Hand assignments followed a blocked schedule. (**D**) Distributions of reaction times (top row) and reach durations (bottom row) for each monkey, hand, and target. Left hand reaches in yellow, right in purple. Horizontal black bars show means, red bars show medians. (**E**) Speed profiles during left- or right-hand trials. Both reaching and stationary hands are plotted in each, although stationary speeds are near 0 and hardly visible. Vertical red lines indicate threshold crossing to mark movement onset. Monkey O main, monkey W inset. Mean +/- standard deviation.

Trials were blocked for each arm, with each block consisting of 2 trials per target in a randomized order (i.e. alternating 2 trials per target for the left arm, then 2 trials per target for the right; Fig 1C).

Average success rates were above 95% for both hands in both monkeys. Overall, reaction times averaged 308 ms for monkey O and 333 ms for monkey W. Distributions of reaction times for each hand/target combination are displayed in Fig 1D, which were fairly consistent across targets. Reach biomechanics varied across the workspace, resulting in slightly different reach durations across targets (Fig 1D). In terms of kinematics, the initial feed-forward portions of reaches were smooth and stereotyped (Fig 1E). There was a very slight but significant increase in the speed of the non-reaching hand between Rest (mean–monkey O: 1.1 mm/s; monkey W: 2.9 mm/s) and Move (mean–monkey O: 3.6 mm/s; monkey W: 7.6 mm/s) phases

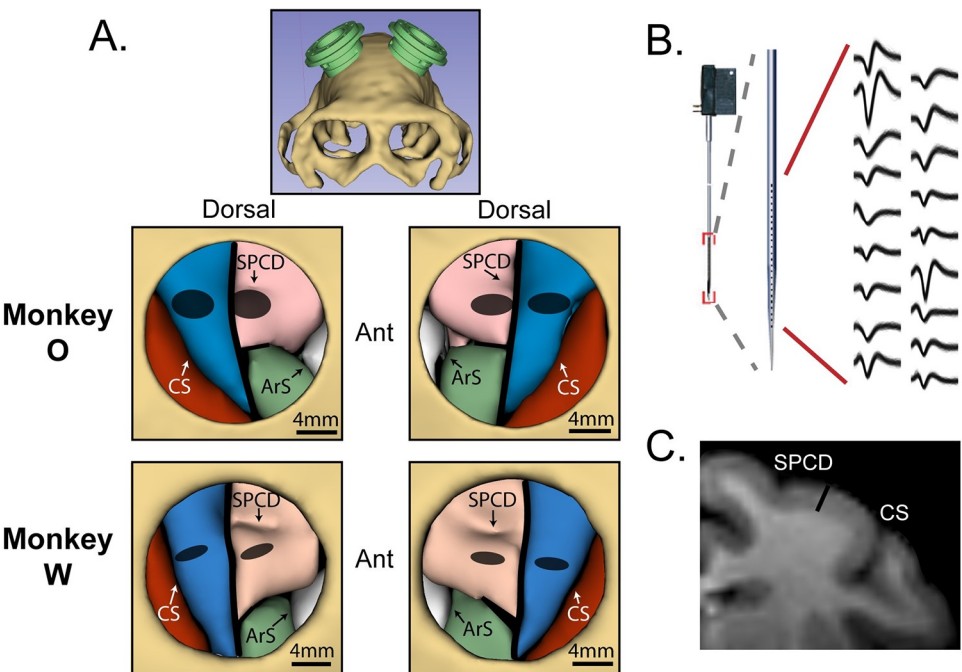

**Fig 2. Neural recordings. (A)** MRI-based volume renderings of the skull and target brain regions. Top panel shows the arrangement of the two chambers. Two bottom rows show segmented brain regions within the cranial window of each chamber, for each monkey. Region boundaries were assigned based on [28]. Red—somatosensory cortex; blue—primary motor cortex (M1); pink—dorsal premotor cortex (PMd); green—ventral premotor cortex; white—frontal eye field. CS—central sulcus; SPCD—superior pre-central dimple; ArS—arcuate sulcus. Grey ellipses indicate regions sampled by recordings. **(B)** Interlaminar recordings were obtained using V- and S- probes (Plexon, Inc., Dallas, TX) with 24–32 electrodes aligned perpendicular to the cortical surface. Example waveforms were all simultaneously recorded from a single probe. **(C)** MRI coronal slice, monkey O. 3mm black bar is approximately equal to the distance spanned by electrodes on 32-channel probes. Same landmark labels as in **(A)**.

of the task (permutation test–monkey O: p = 1.0e-4; monkey W: p = 1.0e-4). We note that the task was designed to mimic natural reaching without the use of physical restraints. As such, we assume the small movements in the non-reaching arm are part of the normal behavioral repertoire occurring during natural unimanual reaching. Nonetheless, we will address any reasonable impacts these small movements may have in our neural analyses.

## Arm-dedicated units emerge across task phases while the overall distribution remains relatively arm-neutral

We recorded 408 and 112 single-units in the caudal aspect of dorsal premotor cortex (PMd) in monkeys O and W, respectively, and 303 and 262 single-units in primary motor cortex (M1) (Fig 2). Since both arms were used in the behavior, we can evaluate the ipsi- and contralateral responses in each unit. Units were pooled across hemispheres in the analysis, with contralateral summaries reflecting the collection of responses during trials performed with the contralateral arm, and vice-versa for trials performed with the ipsilateral arm. PMd and M1 units were analyzed separately. Firing rates were soft-normalized using the Rest phase mean and standard deviation, and modulation strength is expressed as the mean squared value of these standard scores within the window of interest. This modulation metric is essentially variance, and may be thought of as variance for most purposes.

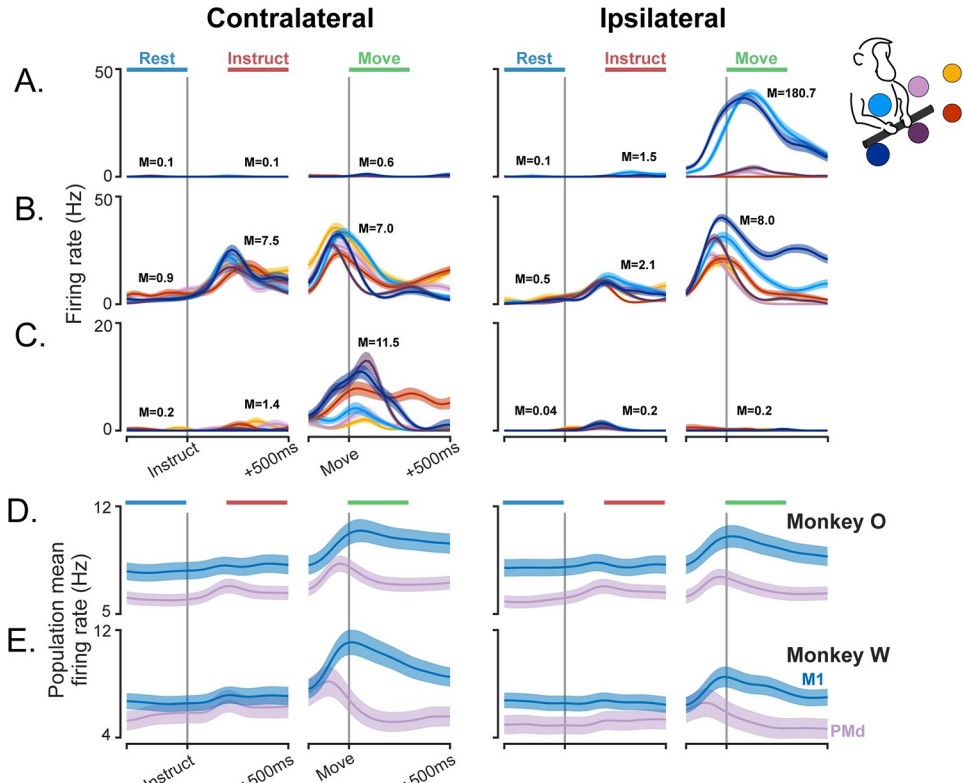

**Fig 3. Firing rate traces of example neurons and population means.** (A-C) Trial-averaged firing rates for three neurons from the left hemisphere. Each line color represents a different target according to the color-coding in the top right. The time windows used to represent each phase in the analysis are indicated by the horizontal bars at the top, and the modulation strength values for each phase are included as annotations (M). Traces display mean +/- SEM. (A) An M1 unit exclusively modulated during ipsilateral movements. Instruct arm preference of -0.84, Move arm preference -0.99. (B) A PMd unit with both Instruct and Move phase modulation for both arms. Instruct arm preference of 0.56, Move arm preference 0.11. (C) A PMd unit with modest contralateral modulation during the Instruct phase and strong contralateral modulation during movement, but no modulation on ipsilateral trials. Instruct arm preference of 0.63, Move arm preference 0.97. (D,E) Mean firing rates for PMd and M1 populations, +/- SEM. The full distributions of modulation and arm preference values for the two populations are provided in Fig 4. These means were calculated over all units and targets; as such, the means reflect the net excitation-inhibition, which is not the same as the modulation strength.

We first analyzed single units to determine the degree of modulation during the Instruct and Move phases of the task (Fig 3). Following instruction, many units in both PMd and M1 became significantly modulated for movements of one or both arms (S1 Table). Units in PMd were, on average, more strongly modulated during the Instruct period than those in M1 (Fig 4A; permutation test, including separate values for each arm–monkey O: p = 8.4e-3; monkey W: p = 7.0e-4). This relationship reversed following movement, with average modulation in M1 becoming stronger than PMd (Fig 4A; permutation test–monkey O: p = 0.015; monkey W: p = 0.018). These results are in line with the view that PMd plays a privileged role in motor preparation. The distributions of modulation values were heavy-tailed and contained some notably extreme values; however, we chose not to apply any outlier criteria. Controls are performed later in our population-level analyses to ensure that results are representative of trends across the entire population rather than a few extreme units.

We next considered the laterality of each unit by quantifying the relative modulation observed during ipsi- and contralateral trials. We expressed each unit's arm preference on a scale from -1 to 1, with 1 indicating exclusive contralateral modulation and -1 indicating

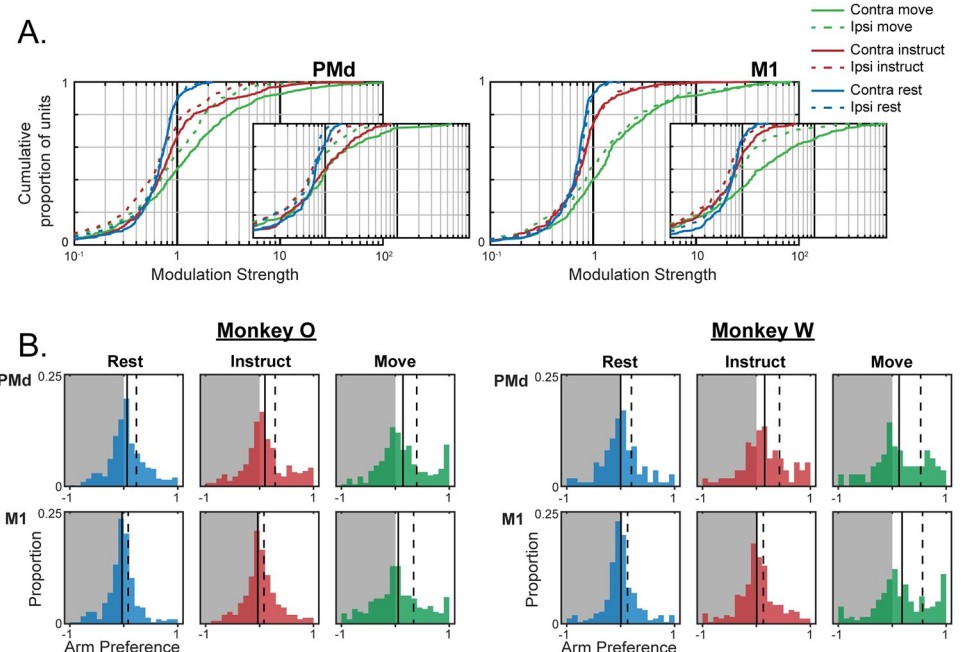

**Fig 4. An increasing number of arm-dedicated units emerge with each task phase.** (**A**) Cumulative distribution of single-unit modulation during each phase, arm. Left panel PMd, right panel M1. Large values cut off by plot: monkey O Contra Move [134(PMd), 133(PMd), 104(PMd)], Ipsi Move [234(M1), 181(M1), 130(M1)]; monkey W Contra Move [125(M1)]. Monkey O main, monkey W inset. (**B**) Distributions of arm preferences during each phase. Negative values are ipsi-preferring (grey background), positive values are contra-preferring (white background). Solid black vertical lines indicate the mean of each distribution, and dashed lines mark the upper quartile.

exclusive ipsilateral modulation (Fig 4B). Although the cue for the forthcoming trial had yet to be presented during the Rest phase, arm selection could be implied from the blocked task structure (Fig 1C). However, except for a very small effect in PMd of monkey O (one-sample t-test–$\mu_{Rest} = 0.06$, p = 2.6e-4), there was no significant contralateral bias observed during the Rest phase in either brain area for both monkeys. Despite the lack of contralateral bias, both monkeys entered arm-specific population states during the Rest phase, which was more pronounced in PMd populations (mean difference between left and right arm firing rates–monkey O PMd: 1.74Hz, M1: 1.59Hz; monkey W PMd: 1.28Hz, M1: 0.92Hz; Fig 4B). For trials in which the same hand was repeated from the previous trial, it was possible to classify the hand for the forthcoming movement from the population activity (S1 Fig). Although the differences in activity during Rest were relatively small when compared to the subsequent task phases, they point to an early specification of the reaching arm that provides context for subspace analyses we present below.

The emergence of laterality after the onset of the instruction cue mirrored the emergence of general unit modulation: A contralateral bias was present in PMd during the Instruct phase and then became present in both PMd and M1 during movement. Mean arm preference in PMd showed a modest but significant bias in the contralateral direction during the Instruct phase (one-sample t-test–monkey O: $\mu_{Instruct} = 0.10$, p = 1.2e-6; monkey W: $\mu_{Instruct} = 0.15$, p = 2.5e-4) and showed no significant change between Instruct and Move (paired-sample t-test–monkey O: $\mu_{Move} = 0.14$, p = 0.07; monkey W: $\mu_{Move} = 0.13$, p = 0.65). Mean arm preference in M1 did not show a significant contralateral bias until the Move phase (one-sample t-test–monkey O: $\mu_{Instruct} = 0.02$, p = 0.20; $\mu_{Move} = 0.05$, p = 0.049; monkey W: $\mu_{Instruct} = 0.02$, p = 0.33; $\mu_{Move} = 0.18$, p = 1.3e-8).

While shifts in the means were modest, changes in arm preference across phases were most evident in the tails of the distribution, corresponding to units that strongly preferred one arm or the other (Fig 4B). These arm-dedicated units typically preferred the contralateral arm, demonstrated by increased occupancy in the contralateral tails of the arm preference distributions; however, a small proportion of the population was exclusively modulated during ipsilateral trials as well (Fig 4B). Despite much of the population remaining arm-neutral (arm preference near 0) or preferring the ipsilateral arm, the emergence of strongly contra-dedicated units was sufficient to drive contralateral shifts in the population mean. In summary, despite much of the population remaining arm-neutral, an increasing number of highly arm-dedicated units emerged with each task phase, primarily favoring the contralateral arm.

## Modulation preferentially occurs within arm-dedicated units

There are two primary means by which population signals can specify the selected arm at each phase. (1) The population may maintain unique covariance structure for each arm that separates signals along different neural dimensions, even if the constituent units are equally modulated for both arms. For example, consider two neurons that are positively correlated for movements of one arm, but negatively correlated for the other arm. When summed, their activity would cancel out in the latter case, whereas it would be amplified in the former. As such, it would provide a signal solely for one arm if the downstream readout were a simple sum (see [11]). (2) Arm-dedicated units may dominate the population response, thereby representing the majority of population variance in dedicated sub-populations. In this case, the arm is simply specified by which sub-population has become active. The latter possibility is investigated over the following two sections. First, we consider whether modulation preferentially occurs in units that are strongly dedicated to one arm or the other.

We performed a regression analysis to quantify the relationship between strength of arm preference and modulation for the preferred arm. Importantly, arm preference and modulation were calculated from independent datasets to prevent artificial linkage between the two measures due to sampling noise. A slope of 1 corresponds to an order of magnitude increase in modulation, on average, when comparing fully arm-neutral units with fully arm-dedicated units. As seen in Fig 5A, the slopes are initially near zero and then become positive over time. To quantify these changes, we used a multi-factorial permutation approach to test for effects of Area (PMd, M1), Phase (Rest, Instruct, Move), and Preferred Arm (Ipsi, Contra) on the population slopes.

We found a main effect of Phase in both animals (monkey O: p = 1.0e-4, monkey W: p = 1.0e-4): a positive correlation between arm preference and modulation strength emerged and strengthened across task phases (Fig 5A and 5B). By the Move phase, there was approximately a ten-fold increase in the modulation strength of units with an absolute arm preference of 1 (completely dedicated) relative to units with an arm preference near 0 (balanced modulation). Since PMd displayed greater modulation than M1 during preparation but not movement, we tested whether the two areas had differing slopes in each phase independently. We found a significant simple effect of Area during the Instruct phase (monkey O: p = 7.0e-4; monkey W: p = 3.9e-3) but not the Move phase (monkey O: p = 0.20; monkey W: p = 0.90). Thus, the relationship was more prominent within PMd prior to movement, while the two areas became roughly equivalent following movement initiation. This was confirmed with a test for 2x2 interaction (monkey O: p = 0.017; monkey W: p = 9.5e-3). Additionally, we analyzed the relationship between arm preference and modulation for the non-preferred arm to confirm that increased arm preference is associated selectively with increased modulation for

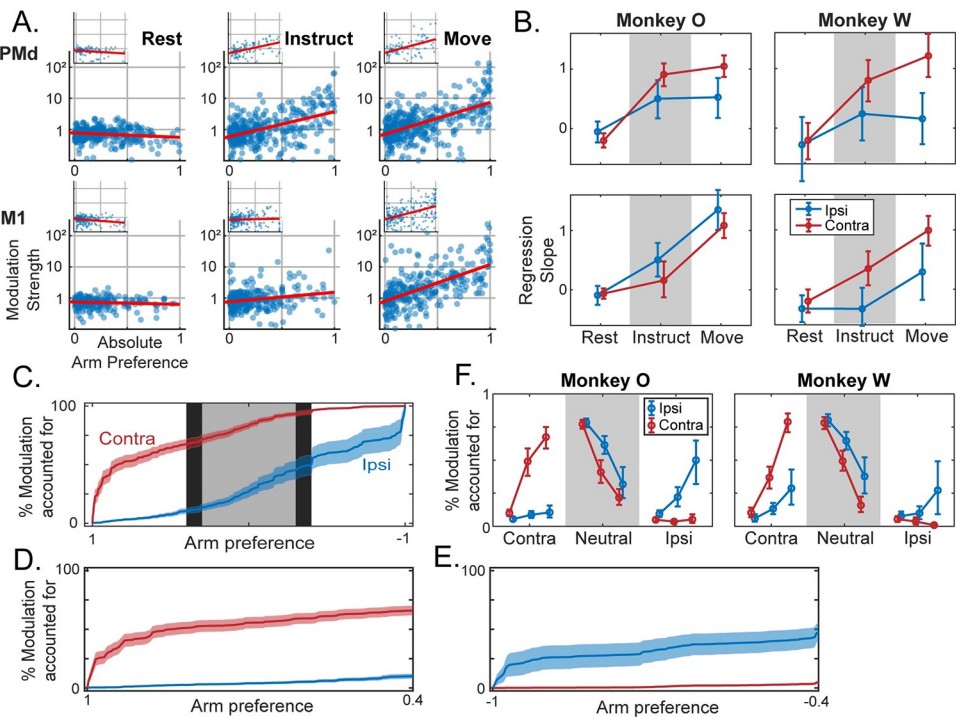

**Fig 5. Neural activity is progressively consolidated within arm-specific subpopulations.** (**A**) Modulation for the preferred arm plotted against arm preference, for all units in each brain area and task phase. Log-linear best fit lines are displayed in red. Inset figures belong to Monkey W. (**B**) Slopes of regression lines fit to data from (A), independently for ipsi- and contra-preferring sub-populations. Mean +/- bootstrapped 95% confidence interval. (**C-E**) For the Move phase in monkey O, cumulative modulation plotted against arm preference, i.e. each point indicates the proportion of modulation accounted for by all units with arm preference values to the left of the indexed position. Positive values on the x-axis indicate contra-preferring, and negative values indicate ipsi-preferring. Shaded error bars indicate bootstrapped standard error. See S2 Fig for cumulative modulation plots for both animals in each phase. (**C**) The full spectrum of arm preferences is shown. Shaded backgrounds indicate three partitions: Contra-dedicated [0.4, 1] and Ipsi-dedicated [-1, -0.4] in white, and Neutral [-0.3, 0.3] in grey. (**D**) Cumulative modulation within contra-dedicated regime. (**E**) Same as (D), but ipsi-dedicated. Note inverted axis. (**F**) The proportion of modulation within each partition from (C) during ipsi- or contralateral movements. Note that the total modulation is significantly lower for ipsilateral movements, particularly for Monkey W, and these data are only displayed as proportions. Mean +/- bootstrapped 95% confidence interval.

the preferred arm (S4 Fig). No significant positive relationships were observed in either monkey, either brain area, or any task phase; therefore, greater arm preference is associated with selective increases in modulation for a single arm.

Given the overall contralateral bias, we further tested whether this relationship held for both contra- and ipsi-preferring units. For the contra-preferring units, there was a significant simple effect of Phase (monkey O: p = 1.0e-4; monkey W: p = 1.0e-4). For the ipsi-preferring units, the Phase effect was significant for monkey O (p = 1.0e-4), but only trended in this direction for monkey W (p = 0.062). Slopes were generally steeper for contra-preferring units; however, the simple effect of Preferred Arm within each Phase was only significant for monkey W (monkey O: $p_{Instruct}$ = 0.13, $p_{Move}$ = 0.93; monkey W: $p_{Instruct}$ = 2.0e-4, $p_{Move}$ = 1.0e-4). It is important to note that monkey W had fewer units with a strong preference for the ipsilateral arm, which makes it more difficult to assess the relationship between arm preference and modulation strength simply due to sampling. Since there are more contra-dedicated units than ipsi-dedicated units for both animals, and those dedicated units tend to be more strongly modulated, these results collectively suggest that a larger proportion of the contralateral signal

exists within dedicated sub-populations compared to the ipsilateral signal. We directly test this conjecture in the following section where we consider population-level implications of these results.

## The population signal is largely confined to arm-specific sub-populations

The preceding analyses establish that there is an increase across task phases in the proportion of units that are strongly dedicated to a single arm, and that those units exhibit greater modulation in activity relative to those that are more neutral. This suggests that the population signal is progressively segregating at the level of individual units. To visualize this segregation, we ordered units based on arm preference and calculated the cumulative modulation at each value, i.e. the proportion of modulation across the entire population that is accounted for by units with arm preferences at or below a certain value (Fig 5C–5E). Since PMd and M1 showed similar relationships in the previous analyses, we combined units from the two areas, analyzing them as a collective population. In the extreme case that population signals are entirely segregated, 100% of ipsilateral modulation would occur at an arm preference of -1, and 100% of contralateral modulation would occur at +1.

We focused on two core questions. (1) Does the proportion of dedicated modulation increase across task phases, indicating a progression towards independent signals? (2) Does the amount of independent (or dedicated) modulation differ for ipsi- and contralateral activation? As expected, dedicated regimes of the arm preference distribution captured a large proportion of the modulation associated with movements of one arm and only a small proportion of the modulation associated with the other arm, primarily during execution (Fig 5C–5F). For statistical testing, we split the arm preference domain into 3 equal width regimes, corresponding to contra-dedicated (arm preference > 0.4), ipsi-dedicated (arm preference < -0.4), and arm-neutral (-0.3 < arm preference < 0.3) units, and summarized the data by expressing the proportion of modulation contained within each regime (Fig 5F). We again used a multi-factorial permutation approach to test for effects of Phase (Rest, Instruct, Move), and Arm (Ipsi, Contra). We will refer to ipsilateral modulation in the ipsi-dedicated units simply as "ipsi-dedicated modulation" and vice-versa for contra-.

For both animals, the effect of Phase was significant in the contralateral responses (monkey O: p = 1.0e-4; monkey W: p = 1.0e-4), with the proportion of contra-dedicated modulation increasing across phases (Fig 5F, red lines). Ipsi-dedicated modulation increased across task phases for both monkeys as well (Fig 5F, blue lines), although this effect was only significant for monkey O (monkey O: p = 1.8e-3; monkey W: p = 0.36). There was a significant interaction between Arm and Phase for both monkeys (monkey O: p = 1.0e-4; monkey W: p = 1.0e-4), indicating the stronger emergence of contra-dedicated modulation as compared to ipsi-dedicated modulation. Both animals showed a simple effect of Hand during the Instruct phase (monkey O: p = 1.0e-4; monkey W: p = 1.0e-4), with more contra-dedicated modulation being observed than ipsi-. This effect was also significant during the Move phase for monkey W (p = 1.0e-4) and approached significance for monkey O (p = 0.053).

These results suggest that arm signals separate at the level of individual units throughout preparation. Moreover, contralateral signals are more independent than ipsilateral signals, in the sense that a larger proportion of the contralateral modulation was represented in dedicated regimes of the population. Since this characterization of the population response captures most of the modulation for each arm in mutually exclusive sub-populations, we will refer to it as the "dedicated" component.

We emphasize that the partitions used here were chosen to broadly isolate the extremes of the distribution; it was not intended that the precise boundaries map onto discrete cell-types

or any similar interpretations. However, we did consider in a post-hoc manner whether the variation in arm preference might be related to some anatomical or physiological factor. To this end, we asked whether units in these different arm preference regimes came from different cortical layers by analyzing their relative depths along the recording probes. The average depth of recorded units did vary across regimes (S3 Fig). Differences in mean depth were statistically significant in M1 during the Move phase for both animals, PMd during the Instruct phase for monkey O, and PMd during the Move phase for monkey W (permutation-based one-way ANOVA–$p < 0.05$). In most cases, this corresponded to the arm-neutral units being recorded at deeper locations than dedicated units, either ipsi- or contra- (see S3 Fig for results of follow-up pairwise tests).

It is possible that the limited range of movement directions used in the task may influence the degree of dedicated modulation (the maximum angle between target vectors is 107˚). For example, a unit that appears dedicated to one arm may only be unmodulated for the other arm over the range of movement directions being tested. That unit may in fact be modulated during different movements, which would cause it to appear neutral if sampled. As a post-hoc control for this possibility, we included data during the return movements following target acquisition, effectively doubling the range of sampled movement directions, and repeated the analyses of Fig 5. The relationship between arm preference and modulation strength remained when analyzing data that included both the reach and return portions of the trial (S5B and S5C Fig). This relationship was significant for both monkeys, in both PMd and M1, and for both ipsi- and contra-preferring units, with one exception (permutation test of regression slopes–$p < 0.05$; monkey W, PMd, Ipsi-preferring units $p = 0.60$). The proportion of dedicated modulation was largely unchanged as well (S5D Fig), and replacing Move phase data with Reach & Return did not impact significance of the Phase effect (Contra-dedicated modulation–monkey O: $p = 1e\text{-}4$; monkey W: $p = 1e\text{-}4$; Ipsi-dedicated modulation–monkey O: $p = 6.5e\text{-}3$; monkey W: $p = 0.26$). Therefore, the dedicated signals that we observe persist even with a broad range of movement directions. Returning to the possibilities outlined at the beginning of the previous section, we therefore conclude that this dedicated component progressively becomes the dominant characterization of the population response–dominant in the sense that it represents the majority of modulation across the population.

## Neural subspaces for the two arms diverge across task phases

We next sought to characterize the time course of changes in neural subspaces as movements were prepared and executed. We hypothesized that dedicated activation would drive population signals into diverging subspaces for the two arms. We identified these subspaces using PCA on simultaneously recorded units (i.e., not trial-averaging activity and combining units across sessions). For these analyses, we pooled units from the left and right hemispheres. We first estimated the dimensionality of the neural subspace during each task phase using a cross-validated data reconstruction method (see Methods; [29]). This is an essential step to avoid drawing conclusions based on noise-dominated dimensions. Dimensionality was calculated separately for each session, arm, and task phase. The dimensionality estimates ranged from approximately 3–5 across the task phases (Fig 6A), values comparable to those reported in previous studies using similar methods [29]. We chose to focus on only four components to represent the neural subspaces of each dataset.

We calculated the alignment between PCA subspaces associated with left or right arm movements using a metric that describes the proportion of low-dimensional variance for one dataset that is captured in the low-dimensional space of another (see Methods; [30]). If the network is organizing activity in the same way across datasets, then the covariance alignment is 1,

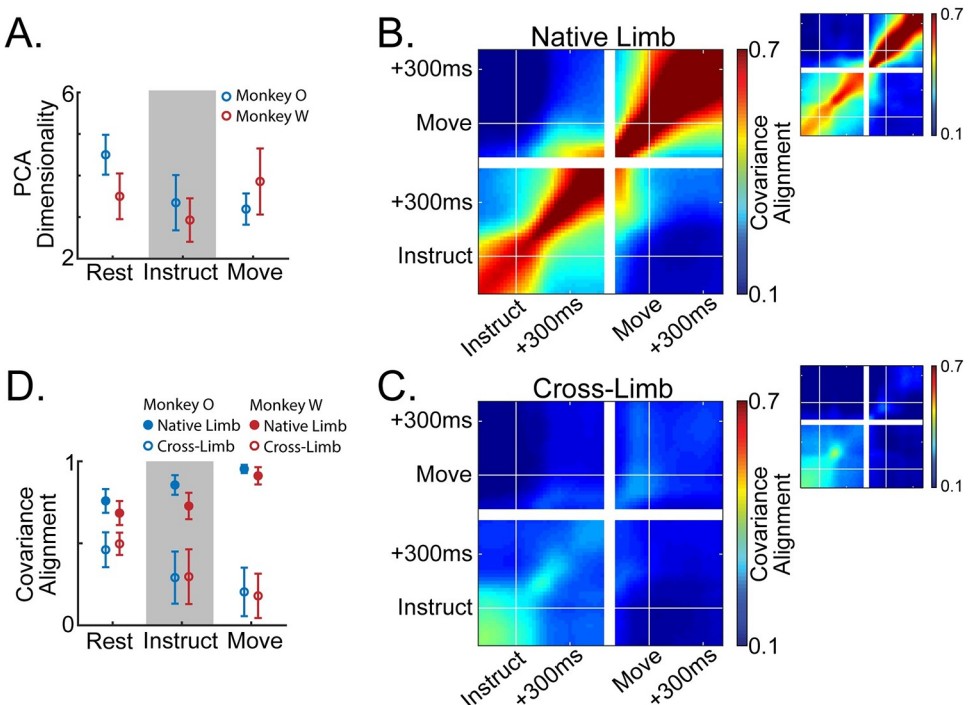

**Fig 6. Population activity reorganizes and diverges for the two limbs throughout planning.** (**A**) Dimensionality of the PCA subspace estimated as the number of components that minimizes the cross-validated reconstruction error of the full-dimensional neural data. Mean +/- standard error across datasets. (**B,C**) Heat maps indicate alignment of 4-dimensional PCA subspaces between all pairs of timepoints across the Instruct and Move phases of the task, averaged across sessions. (**B**) Compares subspaces across time for movements of the same arm. Three blocks forming along the diagonal indicate three distinct subspaces: a pre-instruction Rest space, a post-instruction Instruct space, and a peri-movement Move space. (**C**) Compares subspaces across time for movements of opposite arms. Prior to instruction there is a moderate alignment of the subspaces for each limb, however, the two subspaces diverge around 100ms post instruction. (**D**) Summary of the data in (B,C). Mean +/- standard deviation across datasets.

regardless of signal magnitude. If activity is reorganized into orthogonal subspaces across datasets, then the covariance alignment is 0. Two types of alignment measurements were made: (1) Subspaces were fit to random partitions of trials for the same arm–what we will refer to as "native" alignment–giving us an estimate of natural variability in our subspace estimates when compared over the same time window, and describing the evolution of the motor plan when comparing across time windows. (2) Subspaces were fit separately using trials for either arm and compared with each other–what we will refer to as "cross" alignment–describing the divergence of the subspaces for the two arms at each task phase.

Using single-trial activity event-locked to the onset of instruction and movement, we were able to capture the fine-timescale evolution of any emerging or diverging subspaces (Fig 6B and 6C). When comparing the native alignment across task phases, we observed the emergence of distinct Instruct and Move period subspaces. Fig 6B shows these data displayed as a continuous heat map with block diagonal structure that coincides with the phase transitions. Within each phase native alignment was high, indicating consistent low-dimensional structure in the population activity that was specific to each stage (Fig 6B; Fig 6D filled circles).

As expected, subspaces for the two arms gradually diverged across task phases (Fig 6C; Fig 6D open circles). On the whole, subspaces for the two arms were significantly less aligned than the (cross-validated) comparisons within the same arm (Fig 6D open vs filled circles; two-way

ANOVA, ME comparison type–monkey O: p = 4.0e-68; monkey W: p = 4.2e-32). Interestingly, subspace divergence was already apparent during the Rest phase (paired sample t-test, native-Rest vs cross-Rest–monkey O: p = 6.7e-12; monkey W: p = 6.5e-7). As mentioned in our analysis of single-unit arm preferences, this is likely due to predictable arm assignments from the blocked task structure (Figs 1C and S1). Cross alignment decreased significantly as the trial unfolded, reaching a minimum during movement (one-way repeated measures ANOVA–monkey O: p = 6.2e-8; monkey W: p = 9.5e-8). These results map closely onto the progressive segregation of dedicated signals described in the previous section.

## Subspace separation relies upon dedicated signals

Activity within mutually exclusive sub-populations naturally separates into distinct linear subspaces; as such, we can expect some level of subspace separation as a simple result of dedicated variance. However, it is possible that subspace separation could occur within a distributed representation as well [10,11]. This question is especially important in considering units that show relatively balanced modulation for the two arms. Even though these units show similar levels of activity during contra- and ipsilateral movement, it is possible that their population-level contributions are different for each, and thus also contribute to subspace separation.

To investigate the extent to which subspace separation relied upon dedicated activation, we analyzed the structure of PCA subspaces via their coefficient weights. Since components of PCA models form an orthogonal basis set, each can be independently analyzed to determine its contribution to subspace divergence. We fit separate PCA models for each arm and task phase and calculated two statistics for each component: (1) To capture the contribution of a given component to subspace separation, we calculated the ratio of variance it captured for the two arms (right/left). (2) To capture the dependence of a given component on arm-dedicated units, we calculated a coefficient-weighted average of the arm preferences for all units (e.g., if non-zero weights were only given to right arm dedicated units, this value would be 1; if weights were evenly distributed across the spectrum of arm-preferences, this value would be 0). A strong relationship between these two metrics would suggest that subspace separation relies upon dedicated activation.

Indeed, this was the case during both the Instruct and Move phases. Fig 7A–7C shows a single session example from the Move phase. The top principle components captured a large amount of the variance for the left arm while capturing little variance for the right arm. Components with a variance ratio strongly favoring the left arm almost exclusively weighted units that were themselves highly dedicated to the left arm. The lower components with more balanced variance ratios distributed weights more evenly across the arm preference spectrum. This pattern was evident in each phase throughout recordings from both monkeys. Fig 7D shows the relationship between right/left variance ratio and coefficient-weighted arm preference for the top five principal components of each dataset. Following the instruction cue, components that strongly discriminated between the two limbs (variance ratio far from 1) primarily weighted units that were themselves highly discriminating. This relationship remained strong as the range expanded during the Move phase. The analysis was also repeated using an alternative normalization method to mitigate the effect of highly modulated units. As expected, mitigating the effect of highly modulated units decreased the magnitude of subspace separation while maintaining the relationship between coefficient-weighted arm preference and variance ratio over the reduced range (S6A Fig). This further illustrates the dependence upon highly arm-dedicated, highly modulated units. In summary, these results suggest that the subspace separation described in the previous section relies upon signals that segregate at the level of individual units.

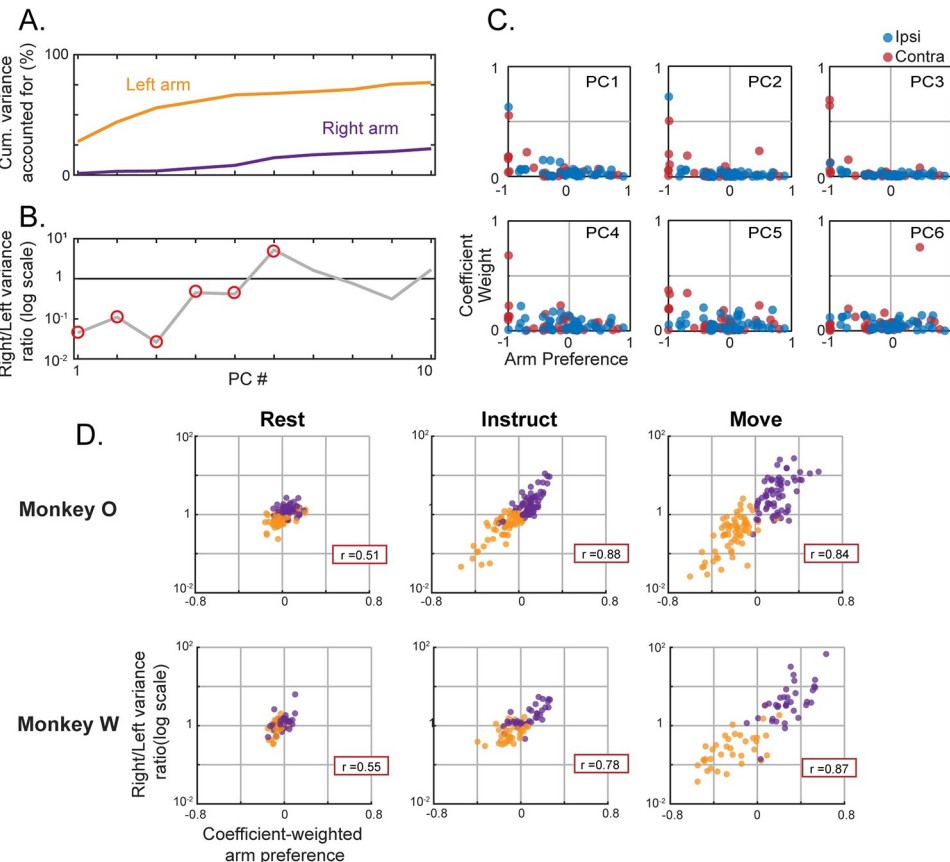

**Fig 7. Separation of arm-specific subspaces relies upon unit-level segregation.** (**A-C**) Single session example of a PCA model trained to capture bi-hemispheric activity during left arm movements. Held-out testing data for 82 simultaneously recorded units were used. (**A**) Cumulative proportion of variance accounted for across the top 10 principal components. (**B**) For each component, the ratio of the explained variance between the two limbs. (**C**) For each component circled in red in 7B, the absolute values of the coefficient weights are plotted against the corresponding unit's arm preference. Top row represents components 1–3; bottom row represents components 4–6. Positive arm preference values indicate right arm preferring units. (**D**) The component variance ratio for the two arms plotted against a coefficient-weighted average of the arm preferences for each unit in that component. Datapoints represent the top 5 principal components of left or right arm trained models across all sessions. Separate models for each phase are plotted in each column. Because these models include activity from both hemispheres, hands are referred to as "left" and "right" as opposed to "ipsi" and "contra". Pearson correlation coefficient for each dataset is displayed in the red box. Top row monkey O, bottom row monkey W.

## An additional distributed signal contains target-specific information about both arms

The preceding sections make clear that the population signal is dominated by a segregated organization. Nonetheless, it is likely that variance associated with the non-preferred arm of each unit also reflects a meaningful population component, albeit one that is much weaker in magnitude. Indeed, many of the units that we recorded in both PMd and M1 were significantly modulated for both arms throughout preparation and movement (S1 Table). To assess the information content and strength of these secondary responses, we divided the entire population of units from both hemispheres and brain areas into two subgroups based on the preferred arm of each unit from a held-out dataset (Fig 8A). If the signals were entirely dedicated to one arm or the other, each subgroup would only contain information about its preferred arm (e.g., a left arm-preferring subgroup would be predictive of left but not right arm

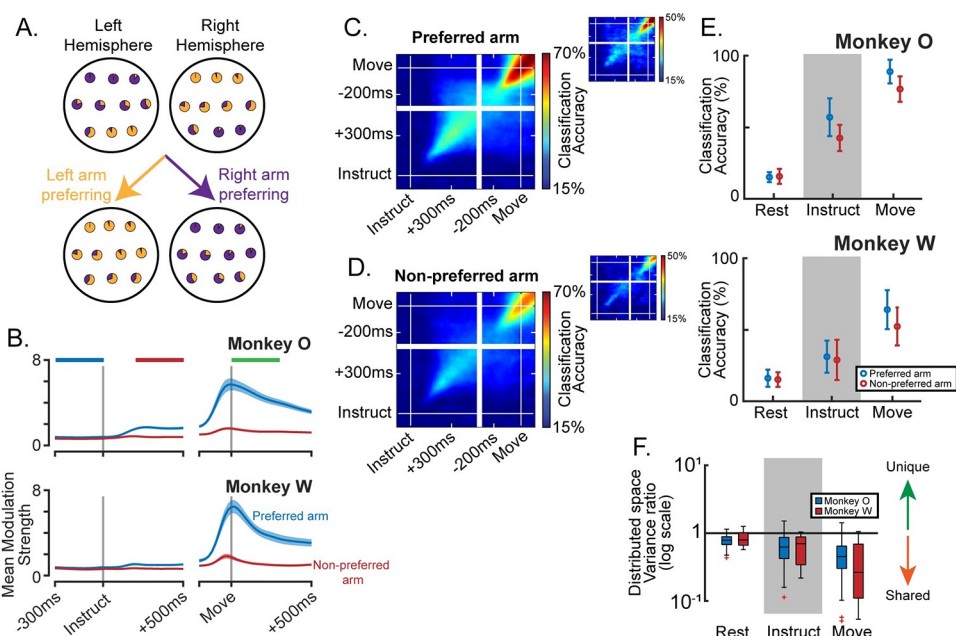

**Fig 8. Behaviorally specific information exists within a subspace that captures bilateral activity.** (**A**) Illustration of the population partitioning approach. Each unit is represented as a pie-chart displaying the relative modulation during left and right arm trials. Most units in the left hemisphere are more strongly modulated during right arm movements (mostly purple pie-charts), yet some prefer left arm movements (mostly yellow pie-charts). Regardless as to which hemisphere each unit is in, the population may be subdivided into left and right arm preferring sub-populations. On the extreme that all information about each arm is contained within dedicated sub-populations, this simple division will fully segregate the signals such that movements of the non-preferred arm cannot be classified. (**B**) Modulation as a function of time, taken as the mean over all units during trials of their preferred or non-preferred arm, +/- standard error. Horizontal bars at the top indicate the phase windows used in analysis. (**C**) Target classification accuracy using LDA for movements of the preferred arm. Models are trained on each time point and tested on each time point to provide high temporal resolution and inform cross-phase generalization of the classifier. Plots are averaged over all sessions (13 Monkey O, large plots, 7 Monkey W, small plots) and both sub-populations (left-preferring, right-preferring). (**D**) Same as (C), but for non-preferred arm movements. (**E**) Summary data of (C,D) for monkey O, top panel, and monkey W, bottom panel. Mean +/- standard deviation across datasets. (**F**) Ratio of the variance captured in the distributed subspace for the two limbs.

movements). If instead there is meaningful activation that is distributed across the same units, then each subgroup would contain both dedicated and distributed information about its preferred arm, but only distributed information about its non-preferred arm.

We first analyzed the time course of modulation for each subgroup during movements of the preferred and non-preferred arms. While modulation during preferred-arm trials was much stronger in the Instruct and Move phases, there was a small amount of modulation during trials of the non-preferred arm as well (Fig 8B). Given the strong directional tuning found in motor cortex neurons (for example, [31]), we assume that modulation associated with the preferred arm carries rich target-specific information. However, it is less clear if activity related to the non-preferred arm carries similar information. To determine whether this modulation carried target-specific information about the behavior, rather than non-specific changes related to task engagement or small movements of the non-selected arm, we trained linear discriminant analysis (LDA) classifiers to predict the target on each trial. Even though the units showed very little modulation when the non-preferred limb was used, prediction accuracy was well above chance (Fig 8C–8E, paired sample t-test with Rest–monkey O: Instruct p = 1.5e-12, Move p = 4.1e-21; monkey W: Instruct p = 1.8e-3, Move p = 1.1e-7). This suggests that the population code is not entirely dedicated but contains a meaningful distributed component as

well. We refer to this as "distributed" in the sense that the contributing units carry information about both arms.

## The distributed signal is contained in a shared subspace for the two arms

We next asked whether subspace separation exists specifically within the distributed portion of population activity. To isolate distributed signals, we again partitioned the population based on preferred arm and fit 4-D PCA models to neural activity during only trials of the non-preferred arm. This is a conservative approach for fitting only the distributed activity, since dedicated activity will be absent during reaches of the non-preferred arm. This approach can also be interpreted as directly isolating the effect of covariance differences by removing the effect of magnitude differences. We will refer to the subspace spanned by these models as the "distributed" subspace.

If population activity for each arm separates along orthogonal neural dimensions, even in the absence of dedicated variance, then the distributed subspace would preferentially capture variance for the non-preferred arm, since that is what it was fit with. Despite having greater magnitude in the ambient space, preferred arm activity would exist largely in the null space of this projection and none of its variance would exist in the distributed subspace. Alternatively, if signals for the non-preferred arm exist within a shared subspace for the two arms, then the patterns of activity for either arm would be preserved through the projection, and we would expect as much or more variance captured for the preferred arm.

Across all task phases and for both animals, more variance was observed in the distributed subspace during preferred arm trials than during non-preferred arm trials (Wilcoxon signed rank–p<0.05 for all six comparisons). The ratios of variance captured for each arm were expressed as non-preferred over preferred and were computed using the raw variance, not the proportion of total variance. Variance ratios were below 1 for nearly every individual dataset and became even lower with each subsequent phase ([Fig 8F]). Additionally, mean coefficient weights were not significantly different across PMd and M1 for the Instruct phase, but were slightly larger for M1 during the Move phase in both monkeys (permutation test–p>0.05 for monkeys O and W Instruct phase; monkey O Move phase p = 1.0e-4, $|\bar{w}_{M1}| = 0.12$, $|\bar{w}_{PMd}| = 0.096$; monkey W Move phase p = 0.027, $|\bar{w}_{M1}| = 0.11$, $|\bar{w}_{PMd}| = 0.096$). This indicates that the two areas were similarly contributing to the effect. Visualizations of the data projected onto the top components in both the dedicated and distributed subspaces provide additional perspective on the temporal evolution of these features, including the relative magnitudes of the signals for each hand and target-specificity ([S7 Fig]). Together these results suggest that across the entire process of preparation and execution of movements, arm signals that are mixed at the level of individual units occupy a shared subspace and are not differentiated through linear population readouts. We again used an alternative firing rate normalization method to confirm that this result was not dependent on overrepresentation of units with the strongest modulation, and the same results were observed ([S6B Fig]). In summary, the subspace capturing distributed activity is not unique to the arm it was fit to, but rather represents a shared subspace for population activity associated with either arm.

## Discussion

We have shown that the combined population response spanning PMd and M1 across hemispheres contains two primary components with regards to the lateralized control of arm movements in non-human primates.

The first is characterized by signals that are segregated at the level of individual units, which we call the "dedicated" component. Activity that emerged following the instructional cue was

most prominent in PMd and showed a tendency for stronger modulation in units with greater arm preference. This caused the signals for each arm to begin segregating into mutually exclusive sub-populations and occupy divergent low-dimensional subspaces. During the transition to movement, M1 became strongly engaged and segregation of arm signals became even more pronounced.

The second component leveraged signals that were mixed within units, which we call the "distributed" component. By splitting the population in two based on each unit's preferred arm and analyzing the responses during non-preferred arm trials, we showed that signals for each arm were not completely segregated. Despite being very small in magnitude, these signals contained target-specific information. In contrast to the natural separability of the dedicated component, however, subspaces fit to this activity captured at least as much variance for the other arm during each phase of the task, suggesting a shared subspace for the two arms that persists across preparation and movement.

## Comparison to previous studies of bilateral arm signals in the motor cortex

Our study adds to a growing body of existing work reporting activity related to both arms in the same motor cortical units during either preparation [6,8,12] or movement [5,7,8,10–12,32]. Two recent studies have addressed the puzzling presence of bilateral activity in M1 during unimanual behavior [10,11]. Despite many units being active for either arm (similar to our own observations in Fig 4B and S1 Table), both studies reported separation of population-level arm signals into distinct neural subspaces. Furthermore, they attributed this separation to covariance changes between the two modes that would cause even signals that are mixed within units to contribute to the effect. In the present study, we build upon these observations and reveal an underlying organizational structure that suggests arm signals are not as mixed within units as they would appear based on distributions of single-unit arm preferences. We show that dedicated signals contribute more to the overall population variance and solely account for the presence of arm-specific subspaces. Signals that are mixed within units reflect a different feature of the population. Our example in Fig 5C–5E demonstrates that segregation of arm signals into distinct neural subspaces likely arises from activation of exclusive sub-populations–similar to the top principal components from [10,11] (as well as those in our Fig 7A and 7B example), dedicated regimes of the population in our study captured large amounts of variance for one arm while capturing very little for the other. Analysis of model structure confirmed that dedicated variance drove PCA results (Figs 7, 8 and S7). Furthermore, we mapped the development of this signal feature across preparation and movement, showing that it begins to emerge even during preparation (Figs 5, 6 and S7) and involves both PMd and M1 (Fig 5).

Not only does this clarify the mechanism of signal separation in motor cortical activity, but it acknowledges a critical heterogeneity in the population response. Dedicated signals represent most, but not all, of the population variance. Signals that are mixed within units (distributed signals) reveal a portion of the population activity that is not independent for the two arms. Effector-independent coding has long been appreciated in PMd [8,12]. Recently, Willet et al. [33] identified separate "limb-coding" and "movement-coding" dimensions in neural activity from the hand knob area of human premotor cortex. The "movement-coding" component represented movements for different effectors in the same fashion. This bears similarity to our "distributed" component, although we only show that activity for each arm resides in the same space, not that its relationship to behavior is invariant. Notably, PMd units were not more heavily weighted in our distributed models. This suggests that even in M1, signals for the two arms are not fully independent.

## Interpretation from a whole-body control perspective

A holistic interpretation of the current results requires considering that the arm is part of a highly coupled biomechanical plant that encompasses the whole body. Under normal reaching conditions, postural muscles not only act to support arm movement, but also serve to stabilize the trunk under reactive forces that arise from the moving limb [34–36]. For the behavior in our study, one would expect that the muscles most directly involved in moving the arm would display the greatest activity changes, namely the muscles of the shoulder girdle, triceps, and biceps. The dedicated component we observed, constituting the majority of the population signal, had a similarly direct and exclusive relationship to the moving arm. In contrast, the distributed component that was characterized by negligible arm-specificity may reflect activity related to more axial musculature associated with posture stabilization during movements of either arm. While we have not recorded EMG activity, the micromovements observed in the non-performing arm (Fig 1E) could reflect forces transmitted across the body. Our results may therefore reflect heterogenous features displayed across all muscles involved in reaching movements, both directly and indirectly, while maintaining a bias towards the arm muscles most directly involved in moving a single arm. Despite recording in regions of the motor cortex that are nominally arm-related, it is plausible that activity related to postural stability via axial musculature is also present in our recordings. The idea that whole-body information is present within nominal arm areas of the cortex has been highlighted by [33].

While the absence of EMG data in the current study limits our ability to directly examine postural-related activity, such recordings have been made in similar contexts (although most of these have been limited to a constrained 2-D workspace, and recordings from trunk muscles have been uncommon). EMG recordings from shoulder and arm muscles in the non-performing arm typically show little or no activity [9–11]. Cisek et al. [8] examined some of the axial musculature during bilateral arm movements while obtaining cortical recordings in the primary and dorsal premotor cortices. The authors found several muscles that displayed activity tuned to the direction of movements for either arm, primarily in the more proximal and axial musculature. If the distributed component identified in the current study is related to axial musculature, this could explain the ability to classify movements of the non-preferred arm (Fig 8D). Notably, Cisek et al. rejected the hypothesis that bilateral instructed delay period activity in PMd was devoted to axial musculature, since it featured effector-independent properties that were not observed in the majority of muscles. However, their reports did not preclude the possibility that activity in M1 or PMd during ipsilateral arm movements may be related to coupled activation of postural muscles in the contralateral body (see also [32]).

A related hypothesis is that the arm-independent distributed component may reflect reflex modulation in anticipation of the postural adjustments that arise during arm movements. Premovement modulation of spinal reflex components has been widely reported, including H-reflexes [37,38], spinal interneurons [39], and even muscle spindles [40] (but see [41]). This hypothesis could explain the presence of distributed activity during the instructed delay period in the present study (Figs 8 and S7) when EMG should be minimal or entirely absent. A likely anatomical substrate for anticipatory reflex modulation is the cortico-reticulospinal pathway given its bilateral projections that have been strongly implicated in postural control [1,42,43] and bilateral coordination of arm movements [44]. Through its connections with alpha and gamma motor neurons it may either directly influence muscle activation or alter musculoskeletal dynamics by modulating reflex gain [45], all in the service of coordinating bilateral axial control with the distal effectors that enable reaching.

While we have dichotomized the dedicated and distributed components of population activity in the motor cortex for analytic purposes, this activity may represent a more fluid

continuum of arm preference that is reflective of the muscles they are involved in controlling. However, although providing postural control is likely a major role of bilateral cortical activity, three points suggest that it serves additional functions. First, ipsilateral activity has been observed, to a lesser extent, during isolated hand and finger movements [12–14]. Such movements are less likely to entail forces that would require postural adjustments involving the contralateral side of the body. Second, postural effects should be most relevant during the Move phase, yet we observe this distributed activity even during the instructed delay. While EMG activity in axial muscles has been reported during the instructed delay period of reaching tasks, that activity is qualitatively different from concurrent neural activity, at least in PMd [8]. Nonetheless, the instructed-delay activity could be related to anticipatory reflex adjustments. Third, we observe a small amount of exclusively ipsilateral neural activity (Figs 3A and 5). This activity seems unlikely to be postural given the absence of evidence for unique muscle patterns on the non-performing side of the body during similar reaching tasks [8].

## Progressive segregation of arm-dedicated signals and its functional significance

To our knowledge, this study is the first to compare low-dimensional population structure during preparation of left vs right arm reaching in neurologically-intact subjects (see [33] for preparation of attempted movements in a human participant with C4 spinal cord injury). It has been proposed that neural subspaces reorganize between preparation and execution of reaching movements [22], which we observe in our own data (Fig 6B). A principle interest of this study was to determine how the emergence of arm-specific signals maps onto this reorganization process. Since previous work has shown that the transition from preparation to movement coincides with an increased proportion of lateralized units [8,17,18], we expected activity to progressively segregate at the level of individual units, represented primarily in the contralateral hemispheres, as the population reorganizes between task phases. This was indeed the case, and began even during the instruction phase (Fig 5). Careful inspection of model structure revealed that segregation of arm signals at the individual unit level drove separation of arm-specific neural subspaces gradually throughout the trial (Figs 6C, 6D and 7). This segregation reached its maximum during movement, suggesting that it was relatively unaffected by the small movements observed during the Move phase (Fig 1).

Importantly, contralateral signals were more independent than ipsilateral ones; a larger proportion of contralateral modulation occurred in contra-dedicated units than the reverse case for ipsi- (Fig 5). This was not a surprising result, as contralateral bias in the functional organization of motor cortex has been clearly revealed by effects stroke [46], lesion studies [2], and cortical stimulation [47–49]. One candidate hypothesis for the presence of ipsilateral activity is that it supplies an independent control signal. There is some evidence that ipsilateral cortex plays an increased role in movement following hemispheric damage [2,50–52], though not necessarily a beneficial or compensatory one. The magnitude of ipsilateral encoding increases with the degree of movement complexity [13] and may involve spatially distinct neural populations [53,54]. However, the corticospinal tract (CST) is almost entirely contralateral, and the effectiveness of the ipsilateral component has been debated [3,55–57]. Ipsilateral cortex may also exert its influence via connections made in the reticular formation, one source of descending bilateral input to the spinal cord [48,52,53,57]. Our results showed a small amount of independent ipsilateral activity (monkey O more so than monkey W), with more of the ipsilateral signal coming from non-dedicated units (Fig 5). Thus, if the ipsilateral hemisphere provides any independent control signal, it is much weaker than the contralateral signal. Rather, our results suggest that ipsilateral signals are involved in some form of bilateral control.

## Bilateral signals and their role in motor control

Correlated cortical activity for movements of the two arms has been widely reported in the literature, primarily using macro-scale neurophysiological approaches. Increases in excitability of homologous effectors during transcranial magnetic stimulation (TMS) [58] and symmetric activation patterns in functional magnetic resonance imaging (fMRI) [14,59] suggest that bilateral motor cortical circuits are organized with mirrored properties. Similar correlated structure has also been reported in human ECoG [60] and premotor spiking activity [33]. Effector-independent activity (correlated in an extrinsic reference frame) has also been observed during movement preparation in PMd [8]. Mirror activation and other forms of interhemispheric communication have been proposed to support intermanual skill transfer [14] or shaping of contralateral activity patterns during complex behavior [13]. However, correlations between the tuning for ipsilateral and contralateral arm movements in M1 units tend to be weak or absent [7,8,11]. In the present study we have not directly compared directional tuning, yet we did observe that the distributed component of bilateral signals existed within a shared subspace for the two arms (Figs 8 and S7). Mirror activity would necessarily reside in the same neural subspace for each arm, provided that the subspace is linear, given that linear subspaces are invariant with respect to reflection. Our results are therefore consistent with functional hypotheses of ipsilateral cortex involving mirror symmetric activation, and more generally for hypotheses that predict linear correlations between activation patterns for the two arms. We note, however, that while consistent linear correlations in the tuning properties of individual neurons would deterministically result in shared neural subspaces, a lack of linear correlation does not mean that neural subspaces will be orthogonal.

The distributed component that we have characterized may also play a role in bimanual coordination. Distinct bimanual activity patterns have been observed in caudal premotor regions using human fMRI [14] and in M1 using single-unit recordings in monkeys [5,7]. Surgical transection of the corpus callosum, the primary direct connection between hemispheres [61], disrupts typical spatial coupling and continuous synchronization of arm movements as well [62,63], suggesting a cortical locus for these forms of bilateral control. These studies suggest that bilaterally distributed networks involving PMd/M1 may facilitate bimanual coordination, a function historically attributed to the supplementary motor area [64]. Our task involved unimanual movements, containing no component of coordination. However, the result that target-specific information existed within a shared subspace (Fig 8) is consistent with a role in coordination. We make limited claims on this hypothesis due to our simplified behavior, and stress that implicating a role in bimanual coordination does not simply mean revealing a shared substrate for signals of both limbs. Nonetheless, a bi-hemispheric network structure may underly computations for controlling the two arms as a unified plant [65]. M1 has been implicated in multi-joint integration for voluntary movement and feedback control [66,67]. Bimanual behaviors have a similar task of overcoming redundant degrees of freedom [68]; many patterns of behavior for each arm independently may help one achieve an action goal so long as cooperation of the two remains intact ("motor equivalence" [69]). This lower-dimensional behavioral coordination space, sometimes called "the uncontrolled manifold" [70], would likely have a similar neural manifold in which bilateral arm signals interact (for related discussion and review, see [71–73]). The distributed space that we report may reflect such a manifold.

## Interpretation from a dynamical systems perspective

One unified explanation for the two components identified in this study is that they represent the computational (or "hidden") layers and the output layer of cortical processing. In this framework, the distributed signal would reflect a bilateral network that plays a supportive role

in motor processing rather than direct output. The idea that bilaterally distributed networks contribute to computations that do not directly represent the output has been previously proposed [10]. Preparatory activity in motor areas reflects abstract features of action and may lack a strong contralateral bias [6,8]. The distinctive lack of laterality in the distributed signal we observed is consistent with other reports of abstract preparatory responses. It played a relatively stronger role during preparation as well, since the dedicated component did not fully develop until movement. This aligns with reports that behaviorally specific features become more apparent in motor cortical signals during active behavior, including laterality [8,16].

From a dynamical systems perspective, distributed signals could serve to enforce internal dynamics of the overall population. Preparatory signals in pre- and primary motor cortex are thought to converge on an ideal population state, or initial condition, such that internal circuit dynamics will guide appropriate patterns of activity for the upcoming movement [24,26,27,74]. Rodent studies have shown that preparatory activity in motor cortical neurons projecting to other cortical areas lacks strong laterality, while neurons with descending output exhibit pronounced contralateral bias and became active closer to movement onset ([17] left vs right directional licking task; [18] left vs right arm pedal pressing task). Furthermore, these bilaterally distributed networks provide robustness to unilateral perturbation during preparation, and it has been hypothesized that the two hemispheres operate together to maintain the network state ([26,27] left vs right directional licking task). The two components that we have identified generally align with this form of network structure. In addition to setting the initial state, persistence of the distributed component during movement may reflect the ongoing dynamics of pattern generation [24,25].

Within this interpretation, progressive segregation of arm signals may reflect the emergence of descending output that mirrors the well-established laterality of anatomical pathways [2,3]. However, descending cortical projections originate primarily in layer V of the cortex, while cells in more superficial layers II and III primarily provide local and interhemispheric connections [75]. We did not observe a bias for arm-dedicated units to be located at deeper recording locations; in fact, the trend was in the opposite direction (S3 Fig). Alternatively, the dedicated component may contain a timing signal for triggering action or transitioning the network from preparation to movement [25,76] while simultaneously specifying the selected effector. Signals that reflect the timing of movements, but not their direction, have been shown to capture the most variance in PMd/M1 population responses [76]. Premotor activity has also been shown to contain "limb-coding" dimensions that specify a movement effector independently of the movement type [33]. The large dedicated signals that we observe bear similarity to both of these previously identified response features, as they capture the majority of population variance and clearly distinguish the chosen effector (however, they also contain information regarding the movement direction). It is plausible that all three reflect the same underlying computational process of effector selection and movement initiation, with each emphasizing a different aspect of the process as a result of the way they were analyzed.

In summary, we present a statistical description of arm signals spanning M1 and PMd throughout reach preparation, characterizing in detail both lateralized and non-lateralized features of the population response. The two components that we have identified will be crucial for contextualizing current theory on bilateral motor cortical processing as well as designing future experiments that investigate the independence and interaction of signals across the hemispheres.

## Methods

### Ethics statement

All procedures were conducted in compliance with the National Institutes of Health Guide for the Care and Use of Laboratory Animals and were approved by the University of California at

Berkeley Institutional Animal Care and Use Committee under protocol ID AUP-2014-09-6720-1.

## Behavioral recordings and task

Kinematic data were collected using LED-based motion tracking of several points along each arm (Phasespace Inc, San Leandro, CA). 3D positions of each LED were sampled at 240Hz. Prior to offline analysis, these positions were smoothed using a cubic spline and smoothing parameter 0.005 (*cspaps* function–MATLAB). The most distal LED, located on the back side of each hand just below the wrist, was used for online endpoint feedback and all offline analysis.

Monkeys were trained to perform a variant of an instructed-delay reaching task (Fig 1B). Endpoint feedback of each arm and all visual stimuli were presented to the animal using a custom-built virtual reality 3D display. This display consisted of two mirrors that projected shifted images independently to each eye to produce stereopsis. Cursors, indicating effector endpoint position, were color coded for the left (yellow) and right (purple) hands, as were all associated stimuli.

Each trial began with the appearance of the start positions for each hand (spherical targets, radius 4cm), located near the body on top of a physical bar that the monkey rested its hands on (Fig 1A). The horizontal positioning of the start targets along the bar fit one of three configurations: both targets 4cm from the body midline, or either hand located 15cm from the midline with the other 4cm from the midline (i.e., both hands centered or one hand eccentric). In a self-initiated manner, the monkey would assume the start position by placing both cursors in their appropriate start targets and maintaining that position for 500 ms ("Rest" phase). Behavioral analysis (Fig 1D and 1E) was performed only on trials where the non-reaching hand was held in the eccentric position.

Our threshold for detecting movement online was 9cm/s; breaking this threshold with either hand during the Rest phase would abort the trial. Marking the beginning of the "Instruct" phase, a cue (spherical target, radius 3cm) would appear at one of six locations within a fronto-parallel plane 8cm in front of the start positions (Fig 1A). For starting configurations in which the non-reaching hand was held in the center position, the lower target on the side of the non-reaching hand was excluded to prevent the reaching hand from coming into close proximity with the non-reaching hand. The color of the cue indicated the required arm, and position of the cue was the target location for the forthcoming reach. The instruction cue remained visible through the delay period, a duration that was sampled uniformly on the interval 500-1500ms. Movement beyond the speed threshold with either hand would abort the trial.

At the end of this period, two simultaneous changes signaled the monkey to move and marked the start of the "Move" phase. First, the sphere defining the start position for the cued arm disappeared. Second, the cue at the target location enlarged (3cm to 4cm radius). The monkey then reached toward the target and once at the terminal location, had to maintain that position for 250ms. To earn a juice reward, the animal had to initiate the reach within 500ms of the onset of the imperative, terminate the movement within the target's circumference, and keep the non-reaching hand stationary for the duration of the trial. To further emphasize that the trial was successful, the target turned green.

300ms windows were used to represent each phase in data analysis. For the Rest phase, we used the final 300ms before the onset of the instruction cue. For the Instruct phase, we used data in the interval between 200ms to 500ms post-cue. For the Move phase, we used the first 300ms following the onset of movement, defined as when speed of the reaching hand exceeded 10cm/s. We used a late window for the Rest phase to avoid any residual activity associated

with moving to the start positions. The steady state neural response was used to position the Instruct phase window; this was reached approximately 200ms after the onset of the instruction cue (see Fig 7B). The Move window was selected to capture peak neural activity associated with movement while including only the feed-forward portion, which typically lasted 250-300ms (Fig 1C, bottom row). Reach durations were calculated as the time between movement onset and the first point where (1) movement speed dropped below 20cm/s, and (2) velocity in the depth direction reached 0.

## Surgical implantation

Two adult male rhesus monkeys (Macaca mulatta) were implanted bilaterally with custom acute recording chambers (Grey Matter Research LLC, Bozeman, MT). Partial craniotomies within the chambers allowed access to the arm regions of dorsal premotor (PMd) and primary motor (M1) cortices in both hemispheres. Localization of target areas was performed using stereotactically aligned structural MRI collected just prior to implantation, alongside a neuroanatomical atlas of the rhesus brain [28].

## Electrophysiology

Unit activity was collected using 24–32 channel multi-site probes (V-probe—Plexon Inc, Dallas, TX), with 15um diameter electrode contacts separated by 100um and positioned axially along a single shank. Probes were lowered deep enough to cover roughly the full laminar structure of cortex (Fig 2B and 2C). The depth of insertion was determined by (1) measurements of the dural surface prior to recording, and (2) presence of spiking activity across all channels. 2 probes were typically inserted in each hemisphere daily and removed at the end of the session, one in PMd and one in M1. A total of 12 insertion points across PMd and M1 of each hemisphere were used across 13 recording sessions in Monkey O, and 6 insertion points across 7 sessions for Monkey W (Fig 2A).

Neural data were recorded using the OmniPlex Neural Recording Data Acquisition System (Plexon Inc, Dallas, TX). Spike sorting was performed offline (Offline Sorter–Plexon Inc, Dallas, TX). Single-unit waveforms were isolated in multi-dimensional feature space (including principal components, non-linear energy, waveform amplitudes) and rejected if either (1) the waveform clusters were not stable over the course of the session, (2) >0.4% of inter-spike-intervals were below 1ms, or (3) they were clearly repeats of a unit identified on an adjacent channel as determined visually by coincident spiking. To fully ensure that units were not double logged, we eliminated one member of any pair of units that had a firing rate correlation above 0.9 and were within two channels of each other. For population level analyses (PCA, LDA), a small number of multi-units were included. A multi-unit was defined by waveform clusters that separated from the noise cluster and were stable over time, but did not quite meet the inter-spike-interval criteria or contained what might be multiple unit clusters that could not be easily separated. For monkey O, the average proportion of multi-units in each single session population sample was 17%, ranging 12–25%. For monkey W, average 20%, ranging 12–32%.

Spiking data were binned in 20ms non-overlapping bins, square-root transformed to stabilize variance, and smoothed with a 50ms gaussian kernel for all analyses [29]. This provided an effective sampling rate of 50Hz, with each sample time aligned to the center of its associated bin. The window edges were included in all analysis windows (e.g. 300ms windows included 16 samples). Both neural and behavioral data were deposited in the Dryad repository: https://doi.org/10.6078/D1FM6S [77].

## Modulation and arm preference metrics

As a time-varying value, modulation was calculated as:

$$M_t = \left(\frac{x_t - \mu_{Rest}}{\sigma_{Rest} + 1}\right)^2 = z_t{}^2, \tag{1}$$

where

$$x_t : \textit{instantaneous firing rate at time t}$$

$$\mu_{Rest} : \textit{mean firing rate during Rest}$$

$$\sigma_{Rest} : \textit{standard deviation during Rest}$$

This unitless metric reflects the deviation from baseline activity, normalized by baseline fluctuations. The constant 1 was added to the denominator for soft-normalization to ensure that units which were silent during rest did not have exploding values and were not overly emphasized in the dataset. Because some units had slightly different activity on left and right arm trials even before instruction, the standard deviation during Rest was calculated separately for each arm and $\sigma_{rest}$ was calculated as the mean of the two.

Single values of modulation representing each phase were obtained using the same 300ms phase windows that were used in behavioral analysis (see Methods –Behavioral recordings and task). Phase-specific modulation data were concatenated across trials into a $(16m \: x \: 1)$ vector, where $m$ is the number of trials and 16 is the number of samples within a 300ms phase window (includes window edges). The mean over this vector provided our scalar estimate of modulation. Note that this is simply a normalized form of variance, which makes the comparison of single-unit and population-level results more straightforward. To make the relationship with variance explicit, modulation can be rewritten as:

$$M_{phase} = \frac{1}{(\sigma_{Rest} + 1)^2} E\left[(X_{phase} - \mu_{Rest})^2\right] \tag{2}$$

Where the expectation on the right is essentially variance using the Rest mean (i.e. the mean squared deviation from the average Rest phase firing rate).

Arm Preference was calculated independently for each phase of the task using the formula:

$$AP_{phase} = \frac{M_{contra,phase} - M_{ipsi,phase}}{M_{contra,phase} + M_{ipsi,phase}} \tag{3}$$

An arm preference of 1 corresponds to a unit that is exclusively modulated during contralateral trials, while an arm preference of -1 is the same for ipsilateral trials. In Figs 7 and S6, and the accompanying analyses, the convention of left arm and right arm was used in place of ipsi and contra. In analyses that used arm preference along with other features, independent datasets were used to calculate each to avoid any artificial coupling due to sampling noise, e.g. modulation and arm preference. The independent datasets consisted of reaches with the same arm, but with the stationary hand held in a different horizontal starting position. Note also that the scaling factor used in the modulation calculation cancels out of the arm preference calculation, making it invariant to the choice of normalization.

## Principal components analysis

Principal components analysis (PCA) was used to identify low-dimensional representations of population activity with the *pca* function in MATLAB. PCA computes an orthogonal basis set that reflects the principal axes of variation in the data. Individual components do not strictly correspond to observed activity patterns, and one should be wary of interpreting them as such, yet the low-dimensional space spanned by the top few components has been frequently used in systems neuroscience as a helpful descriptor of coordinated ensemble activity [19]. PCA was selected over other dimensionality reduction techniques for its widespread use and relative lack of assumptions. Additionally, PCA was used in two recent papers covering similar topics to this one [10,11]. Therefore, using PCA over other alternatives was also intended to improve generalization of our results to the existing literature.

Prior to fitting the models, firing rate data were soft-normalized using the same method as in the modulation strength calculation:

$$z_t = \frac{x_t - \mu_{Rest}}{\sigma_{Rest} + 1} \qquad (4)$$

An alternative normalization factor was used to create S6 Fig, replacing the denominator by the full firing rate range + 5Hz [10,11,22].

Since Rest phase mean activity was already subtracted from individual units, we did not de-mean again prior to computing PCA models. Measures of variance accounted for were not inflated by capturing means because they were computed using the variance of the component scores (Figs 7 and 8F):

$$V = Tr(Cov(XP)) \qquad (5)$$

Where *X* is a (16*m x n*) data matrix of concatenated trials and *P* is an (*n x p*) projection matrix, with *m* trials, 16 samples in the phase window of each trial (includes window edges), *n* units, and *p* principal component dimensions.

Cross-validation approaches were used for all analyses and figures to address overfitting. This provided accurate and generalizable estimates of variance capturing metrics that could also be appropriately compared across datasets (i.e. across time or arms). For the covariance alignment reported in Fig 6, models were trained on random partitions of the data which included all three starting configurations, and alignment was computed across the two partitions (Monte Carlo cross-validation). The use of all three configurations could partially align the PCA subspaces with the small neural changes associated with different postures. This would be most relevant during the Rest phase, where the most reasonable impact this could have would be causing the subspaces for each arm to appear more aligned. However, we observed the opposite pattern, that the subspaces already differed during the Rest phase (Fig 6), suggesting that this did not make a meaningful impact.

For the analyses presented in Figs 7, 8, S6 and S7, PCA models were tested on datasets that were held out from model training. In these cases, the stationary hand was held in different starting positions for the training and testing sets. While the position of the stationary hand may influence neural activity and its relationship to behavior (i.e., tuning), the subspace spanned by neural activity in either posture should be largely the same and appropriate for cross-validation in PCA. Nonetheless, all analyses were designed such that any generalization cost would be symmetric for the tested conditions and thus unlikely to introduce substantial bias into statistical testing or data visualization.

## Dimensionality estimation

Dimensionality of the PCA subspace was estimated by optimizing the cross-validated reconstruction of full-dimensional neural data from component scores. Only data with both hands in the centered start positions were used. Given $n$ units, $m$ trials, and 16 samples (includes window edges) within the phase window for each trial, the following procedure was used:

1. Leave out the $i^{th}$ trial (16 samples) from the data matrix, yielding training data, $X^{(-i)} \in \mathbb{R}^{16(m-1) \times n}$, and testing data, $X^{(i)} \in \mathbb{R}^{16 \times n}$.

2. Train PCA model of dimension $p < n$ on $X^{(-i)}$, using singular value decomposition (SVD) to compute the projection matrix, $P^{(-i)} \in \mathbb{R}^{n \times p}$

3. Leave out the $j^{th}$ unit from the testing data and projection matrix by removing the $j^{th}$ column and row from each, respectively, yielding $X^{(i)}_{-j} \in \mathbb{R}^{16m \times (n-1)}$ and $P^{(-i)}_{-j} \in \mathbb{R}^{(n-1) \times p}$. This is the current unit that will be reconstructed.

4. Using the Moore-Penrose pseudoinverse, find a new projection matrix with the $j^{th}$ unit removed, whose transpose is $(P^{(-i)}_{-j})^+ \in \mathbb{R}^{p \times (n-1)}$. This matrix projects the $(n-1)$ dimensional neural activity into the original $p$ dimensional PC space, therefore computing component scores in the absence of unit $j$.

5. Calculate the component score for the $i^{th}$ trial using the remaining units and the new projection matrix, then estimate the $j^{th}$ unit from that component score by projecting back into the ambient space. As a single step, this calculation is:

$$\hat{X}^{(i)}_j = [P^{(-i)}(P^{(-i)}_{-j})^+(X^{(i)}_{-j})^T]_j \tag{6}$$

6. Repeat for trials $i = 1,\ldots,m$

7. Repeat for units $j = 1,\ldots,n$

8. Repeat for component numbers $p = 1,\ldots,10$

9. Take the number of components that minimizes the predicted residual error sum of squares (PRESS) statistic:

$$PRESS = \sum_{i=1}^{m} \sum_{j=1}^{n} (X^{(i)}_j - \hat{X}^{(i)}_j)^2 \tag{7}$$

This method reconstructs the full-dimensional neural data, independent of the training set, by identifying consistent population structure. There are no mathematical constraints favoring increased dimensionality, i.e. it is robust to overfitting. As such, the number of components that minimizes the reconstruction error provides a conservative estimate of the dimensions that meaningfully reflect population structure. Similar methods have been used previously for assessing dimensionality reduction techniques for neural data and yielded comparable values [29]. Using heuristics, such as the number of components to explain 90% variance, would be inappropriate for our analyses. They are prone to overfitting, which could include noisy components and impair analysis of model structure via coefficient weights.

Other studies have reported greater dimensionality estimates in PMd and M1, with values ranging from 4 to 16 [78,79]. Estimates across the literature vary in terms of the estimation

method used (cross-validated data reconstruction, variance accounted for), the particular dimensionality reduction model (PCA, factor analysis), and the details of the behavioral paradigm. The cross-validated data reconstruction technique used in the present study will tend to yield conservatively low dimensionality estimates, and PCA also tends to reach optimal data reconstruction in fewer dimensions than other models [29]. It is also possible that our recording electrodes, which were arranged perpendicularly to the cortical surface, may have sampled a restricted subspace of the intrinsic neural manifold as compared to chronically implanted arrays that sample a wider horizontal spread. Neurons in the motor cortex with similar preferred movement directions tend to be organized into columns [80], so each probe may sample units with similar response properties along its axis. Indeed, many of the studies reporting greater dimensionality estimates in the motor cortex have used arrays with more horizontal spread.

## Covariance alignment

We computed a measure of similarity between pairs of subspaces that we call Covariance Alignment. Our method is essentially the same as that previously used for comparing low-dimensional spaces via factor analysis [30]. In short, this measure computes the proportion of low-dimensional variance from one dataset that is also captured in the low-dimensional space of another dataset.

Given data matrices $X_A, X_B \in \mathbb{R}^{16m \times n}$, the following procedure was used:

1. Train PCA models of dimension $p < n$ on $X_A$ and $X_B$, using SVD to compute the projection matrices $P_A, P_B \in \mathbb{R}^{n \times p}$

2. Project $X_A$ into its own $p$-dimensional space and compute the variance as:

$$V_A = Tr(Cov(X_A P_A)) = Tr(Cov(T_A)) \tag{8}$$

3. Project the $p$-dimensional representation of $X_A$, which is $T_A$, into the $p$-dimensional space identified using $X_B$ and compute the variance as:

$$V_{A\_in\_B} = Tr(Cov(X_A P_A P_A^T P_B)) = Tr(Cov(T_A P_A^T P_B)) \tag{9}$$

4. Return the proportion of $p$-dimensional variance from dataset $A$ that is also captured in dataset $B$'s subspace using the ratio:

$$CA = \frac{V_{A\_in\_B}}{V_A} = \frac{Tr(Cov(X_A P_A P_A^T P_B))}{Tr(Cov(X_A P_A))} = \frac{Tr(Cov(T_A P_A^T P_B))}{Tr(Cov(T_A))} \tag{10}$$

This metric is subtly different from the alignment indices used in [10,11,22]. The key difference here is the double projection in the numerator, which means that we are specifically capturing the proportion of low-dimensional variance from one dataset that is captured in the low-dimensional space of another, rather than the ratio of overall variance captured in two different subspaces.

## PCA coefficient analysis

Since components of PCA models form an orthogonal basis set, each was independently analyzed to determine its contribution to subspace divergence. Two statistics were calculated for each component using held-out datasets.

First, we projected activity during trials of each arm onto a single component, calculated the variance of the projections for each arm, and expressed them as a ratio. This captured each component's contribution to discrimination between the arms. For component *C*, this calculation is:

$$V_{C,R/L} = \frac{Var(X_R P_C)}{Var(X_L P_C)} \tag{11}$$

Where $X_R, X_L \in \mathbb{R}^{16\ m\ x\ n}$ are data matrices for the right and left arms, respectively, and $P_C \in \mathbb{R}^{n\ x\ 1}$ is the projection matrix for component *C*. The log of this ratio will be far from 0 if there is much more variance for one arm than the other along the axis defined by $P_C$.

Second, we calculated a coefficient-weighted average of the arm preferences for all units. If non-zero weights were only given to right arm dedicated units, this value would be 1; if weights were evenly distributed across the spectrum of arm-preferences, this value would be 0. Therefore, this measure captured the dependence of a given component on arm-dedicated units. The coefficient-weighted arm preference, *CAP*, for component C was calculated as

$$CAP_C = \frac{A|P_C|}{\sum_{i=1}^{n} |P_{C,i}|} \tag{12}$$

Where $A \in \mathbb{R}^{1\ x\ n}$ is the vector of arm preferences for each unit.

## Linear discriminant analysis

Population coding of movement was analyzed using Linear Discriminant Analysis (LDA) with the *fitdiscr* function in MATLAB. LDA assumes that each class (target x limb combination) is associated with a multivariate normal distribution over the predictor variables (spiking activity of multiple units) having identical covariance but different means.

The feature matrix $X_{LDA} \in \mathbb{R}^{m\ x\ n}$ consisted of a single sample per trial for each of the *n* units. For fine timescale analysis, this was the instantaneous firing rate. For models representing an entire phase, this was the mean firing rate during the 300ms phase window. Uniform priors were enforced for all models. As it was expected that the covariance may change across use of the two arms during reaching, LDA models were trained separately for each limb to allow fitting of arm-specific covariance matrices. The target classification presented in Fig 8C and 8D was performed only on trials where the non-reaching hand was held in the eccentric position. The supplementary analyses presented in S1 Fig included data from all three starting configurations. Targets in different configurations were treated as separate classes to avoid confounds related to the different configurations. LDA was chosen for its robustness to violations of the given assumptions and its history of success with neural data [14,81].

## Fine timescale analysis of population coding and subspace development (heatmaps)

The same basic method was used for displaying fine timescale changes in population coding of movements (via LDA) and covariance structure (via PCA, Covariance Alignment). The method is depicted schematically in S8 Fig. Neural data were organized as 3D tensors (units, time windows, trials). Comparisons were made between all possible pairs of time windows, using fully independent trial sets to prevent overfitting. For LDA models, this consisted of leave-one-out cross-validation; for Covariance Alignment, random partitioning into two data-sets of equal trial numbers. Averages of the cross-validated results provided the 2D matrices visualized using heatmaps in Figs 6B, 6C, 8C and 8D. A single row or column therefore reflects

the similarity of population coding or covariance between a single timepoint and all other timepoints across the trial. Block diagonal structure in the heatmaps reveals locally consistent structure within task phases.

## Permutation testing procedures

Permutation tests were used for both single and multi-factorial hypothesis testing when parametric tests were inappropriate. Null distributions were constructed by constraining permutations to only data that were exchangeable under the null hypothesis [82]. For example, we maintained the crossed structure of Phase (Rest, Instruct, Move), by only permuting Phase labels within units. 10,000 permutations were used for all analyses, and p-values were estimated as the proportion of permutations resulting in test statistics that were at least as extreme as what was observed. In cases where the observed test statistic was more extreme than any permutations, we assigned a p-value of 1/number of permutations = 1.0e-4.

## Supporting information

**S1 Table. Proportions of significantly modulated single-units across task phases.** For well isolated single-units in each brain area, the proportions of the total population that were significantly modulated when compared with the Rest phase (two-sample t-test, p<0.05) are displayed in each cell. For each phase, single-units were classified as uniquely ipsi, contra, or bilaterally modulated. Top row in each pair of rows represents Monkey O, bottom row Monkey W.
(TIF)

**S1 Fig. Arm-specific neural patterns exist during Rest on predictable trials.** Cross-validated classification accuracy for hand (left column) and target (right column) assignments. LDA models were trained on only trials that required use of the same arm as the previous trial, then tested on either held-out repeating arm trials (blue lines) or switching arm trials (red lines). Separate models were used for each timepoint. Horizontal grey lines indicate chance level. 13 Sessions for monkey O (top row); 7 sessions for monkey W (bottom row). Mean +/- standard error across sessions.
(TIF)

**S2 Fig. Distributions of modulation captured by units with different arm preferences.** The cumulative modulation captured by units at each value of arm preference is plotted for both monkeys in each task phase. Units were first sorted according to their arm preference in an independent dataset. The cumulative modulation of ipsi- and contralateral responses was then computed at each arm preference value. Each datapoint indicates the proportion of modulation accounted for by all units with arm preference values to the left of the indexed position. Mean +/- bootstrapped standard error.
(TIF)

**S3 Fig. Relationship between unit depth and arm preference.** (**A**). The depth of the most superficial electrode was set to a value of 0, providing a reference point for the depth of all units (electrodes were spaced by 100um in a line along the length of the probe). Depth of each unit is plotted as a function of arm preference for PMd and M1 in the Instruct and Move phases. The histograms to the right display the marginal distribution of all unit depths. Recordings in Monkey W were done with 24 channel probes, except for the left M1 probe which had 32 channels. This resulted in less sampling between depths of 2.3 and 3.1mm. (**B**). Units were classified as Contra-dedicated, Neutral, or Ipsi-dedicated (see Fig 5C and 5F). For each category, horizontal black lines, circles, and extended vertical lines indicate the mean,

median, and interquartile range. Asterisks indicate significant pairwise differences (permutation test, p<0.05) determined after an initial ANOVA for group differences.
(TIF)

**S4 Fig. Modulation for the non-preferred arm does not increase with greater arm preference.** Companion figure for Fig 5A and 5B. (**A**) Modulation for the non-preferred arm plotted against arm preference, for all units in each brain area and task phase. Log-linear best fit lines are displayed in red. Inset figures belong to Monkey W. (**B**) Slopes of regression lines fit to data from (A), independently for ipsi- and contra-preferring sub-populations. Mean +/- bootstrapped 95% confidence interval. Note the different y-axis from Fig 4B. The slope was not significantly greater than 0 in any condition, meaning that increased arm preference is associated uniquely with greater modulation in the preferred arm, as opposed to an increase for both arms that is just larger for the preferred arm.
(TIF)

**S5 Fig. Dedicated signals persist with increased kinematic range.** To determine whether having a limited range of reach directions was responsible for the observation of arm-dedicated signals, select analyses were performed again on data that included return movements. By including these movements, which were opposite the direction of the outward reaches used in the primary analyses, the range of sampled behavior was greatly increased. (A) Speed profiles for return movements following target acquisition during left- or right-hand trials. Individual trials were aligned to peak return speed, indicated by the vertical red line. Both reaching and stationary hands are plotted in each. Despite being unconstrained by the task, the non-selected hand remained still during the return. Monkey O main, monkey W inset. Mean +/- standard deviation. (B-D) Analyses from Fig 5 repeated using Move phase data concatenated with 300ms of data beginning 200ms before the point of peak return speed, i.e. reach and return. (B) Compare to Fig 5A. Modulation for the preferred arm plotted against arm preference, for all units in each brain area. Log-linear best fit lines are displayed in red. Inset figures belong to Monkey W. (C) Compare to Fig 5B. Slopes of regression lines fit to data from (B), independently for ipsi- and contra-preferring sub-populations. Mean +/- bootstrapped 95% confidence interval. (D) Compare to Fig 5F. The proportion of modulation within each partition from (C) during ipsi- or contralateral movements. Note that the total modulation is significantly lower for ipsilateral movements, particularly for Monkey W, and these data are only displayed as proportions. Mean +/- bootstrapped 95% confidence interval.
(TIF)

**S6 Fig. Subspace analysis using alternative firing rate normalization.** Prior to performing PCA, an alternative method of normalizing firing rates was used for these plots. Rather than dividing by the standard deviation at Rest, each unit's firing rate trace was divided by the full firing rate range + 5Hz [10,11,22]. This will mitigate the effect of highly modulated units, which PCA will preferentially represent otherwise. (A) Repetition of Fig 7D. (B) Repetition of Fig 8F.
(TIF)

**S7 Fig. Dedicated and distributed subspace projections.** A-B. Projections for monkey O. A. 3D projections of activity from neural sub-populations partitioned based on their preferred arm (see Fig 8A). Each PCA model was trained using trials where the reaching hand was either the preferred hand (dedicated subspace) or the non-preferred hand (distributed subspace) of the sub-population. Separate models were trained during the Instruct phase (300ms before to 500ms after instruction onset) and the Move phase (200ms before to 500ms after instruction onset). The models were trained on trial-averaged data for each target to provide a single

visualization for each group. Each target trace is color coded according to the cartoon in the upper right. The projected data is from an independent validation set that included only 5 of the 6 targets. Solid lines indicate data using the same hand as the training set (native hand), and dashed lines indicate the opposite hand projections (cross hand). B. The data from A plotted against time for the top 4 PC's. C-D. Projections for monkey W.
(TIF)

**S8 Fig. Method for fine timescale analysis of population coding and subspace separation.**
This schematic outlines the process for fine timescale analysis of population coding using LDA and leave-one-out cross-validation. Neural data were organized as 3D tensors (units, time windows, trials). Models were trained to predict targets using a single time window and all but one trial. Those models were then used to predict the target on the held-out trial, making separate predictions based on neural data from each time window. The process was then repeated using the next time window as training data until all possible pairs of time windows had been used as training and testing data. This constituted a 2D matrix of "hit" booleans (number time windows x number time windows) for the predictions of a single trial. After iterating over all trials to be used as held-out test data, the mean was taken across trials to construct a single 2D matrix of classification accuracy. The same basic process was used for visualizing the development of subspace separation, but instead of leave-one-out cross-validation trial sets were repeatedly divided into two random halves of equal size. Covariance alignment was then computed between all possible pairs of timepoints for the two disjoint trial sets.
(TIF)

## Acknowledgments

We thank M. Kitano for help in NHP care and handling, and A. You, E. Formento, W. Liberti and Z. Balewski for methodological advice and feedback.

## Author Contributions

**Conceptualization:** Tanner C. Dixon, Christina M. Merrick, Richard B. Ivry, Jose M. Carmena.

**Data curation:** Tanner C. Dixon.

**Formal analysis:** Tanner C. Dixon.

**Funding acquisition:** Jose M. Carmena.

**Investigation:** Tanner C. Dixon.

**Methodology:** Tanner C. Dixon.

**Project administration:** Jose M. Carmena.

**Resources:** Joni D. Wallis, Jose M. Carmena.

**Software:** Tanner C. Dixon.

**Supervision:** Richard B. Ivry, Jose M. Carmena.

**Visualization:** Tanner C. Dixon.

**Writing – original draft:** Tanner C. Dixon.

**Writing – review & editing:** Tanner C. Dixon, Christina M. Merrick, Joni D. Wallis, Richard B. Ivry, Jose M. Carmena.

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
