## [Decision Letter · Decision Letter 0]

11 May 2021

Dear Mr. Dixon,

Thank you very much for submitting your manuscript "Hybrid dedicated and distributed coding in PMd/M1 provides separation and interaction of bilateral arm signals" for consideration at PLOS Computational Biology.

As with all papers reviewed by the journal, your manuscript was reviewed by members of the editorial board and by several independent reviewers. In light of the reviews (below this email), we would like to invite the resubmission of a significantly-revised version that takes into account the reviewers' comments. I would like to draw your attention that the main issue raised by actually both reviewers is that they prompt a more careful thinking about the results in the context of their biological complexity. We believe tackling this challenge will help not only greatly improve the manuscript but also the impact of your work.

We cannot make any decision about publication until we have seen the revised manuscript and your response to the reviewers' comments. Your revised manuscript is also likely to be sent to reviewers for further evaluation.

Sincerely,

Aldo A Faisal

Associate Editor

PLOS Computational Biology

Daniele Marinazzo

Deputy Editor

PLOS Computational Biology

Reviewer's Responses to Questions

**Comments to the Authors:**

Reviewer #1: Summary:

Monkeys made unimanual movements of the contralateral or ipsilateral arm to 6 targets (two horizontal rows of targets at the Left, Midline, and Right, one above the other), in a pseudo-randomized-block sequence in an instructed-delay task paradigm. Multiple sets of simultaneous neural recordings were made in M1 and PMd using multi-port harpoon electrodes over several daily recording sessions. Neural data were subjected to quantitative analyses including PCA-based dimensionality reduction, covariance alignment and linear discriminant analysis. The authors identified two main discharge components that they called “dedicated” and “distributed”. The “dedicated” component accounted for most of the variance, segregated activity into neurons that were mainly contralateral- or ipsilateral-arm preferring, emerged first in PMd during the Instruction period, increased further during movement, and was stronger in the contralateral hemisphere. In contrast, the “distributed” component was much weaker, did not discriminate the laterality of single-neuron activity at any phase in the trial, and was distributed across both hemispheres. The results suggest two independent functional layers or modes of operation of the motor cortical neural network. The authors proposed that the “distributed” component might be more implicated in bilateral preparatory computations and in ongoing internal processing within the network that performs certain computations that are not directly related to output control signals, while the “dedicated” component might be more implicated in generating unilateral output signals in the hemisphere contralateral to the moving arm.

General comments:

This study looked for population-level neural correlates of the control of contralateral versus ipsilateral arm movements in an analytic space of distributed patterns of neural discharge variance and covariance, and in extracted orthogonal dimensions of variances in a PC space. The neural data and analyses are a novel and valuable addition to the literature on the control of arm movements. However, where the manuscript falls short in my opinion is when it interprets the findings in terms of a control problem in a nearly equally abstract and frankly simplistic “engineering” space of ipsilateral versus contralateral arms and target locations/trajectories. It fails to take into account that those neural circuits are trying to implement those actions by controlling a complicated non-linear skeletomotor plant. The authors need to think through the control problem in more biologically- and anatomically-correct terms. The following is a disorganized set of comments about this very interesting study.

Specific comments:

1) Somatotopy still matters: The neural data were collected in nominally “arm-related” regions of M1 and PMd (Figure 2), but it is entirely likely that the neurons recorded in any given day and across days were a diverse group bearing different degrees of coupling to motor outputs of different parts of the arm and body, ranging from the wrist and hand, to the shoulder, shoulder girdle and axial structures. The Willet et al. (2020) study brings this issue into clear focus. The authors did not attempt to characterize the output relationships of each recorded neuron. I’m not criticizing them for that, because I can imagine how challenging that would be. However, this issue is not raised at all in the interpretation of their findings. If a given neuron shows a clear preference to contralateral over ipsilateral arm movements or vice versa, the functional significance of that response property will surely be different if it turns out that it is primarily coupled to wrist movements in one arm that are unlikely to be evoked during movements of the other arm, or to shoulder or axial structures that are much more likely to be activated during movements of both arms (see next comment). Likewise, if their contra-preferring versus ipsi-preferring neurons were non-uniformly biased to more distal versus more axial body structures, respectively, that would have a significant impact on the functional implications of the response differences that they documented. The manuscript looks for patterns in an abstract space of neural discharge variances without taking into account that different neural sub-populations are very likely primarily controlling the actions of different parts of a complex skeletomotor plant, beyond the dichotomy between contralateral and ipsilateral arms. The authors need to address the limitations of interpretation resulting from this uncertainty in the nature of their neural data sets.

2) The (ground) truth is out there: It resides in biomechanics and muscle activity. This follows up on the previous comment. The authors are confronted with the challenge of interpreting the functional meaning of neural activity evoked while the monkeys performed movements using the arm contra- or ipsilateral to the recording site. They conclude that their results “suggest that ipsilateral signals are involved in some form of bilateral control” (lines 632-633) and speculate about how this might contribute to the control of the ipsilateral arm by “mirror symmetric activation” (line 651) and/or in bimanual coordination (lines 636-677). However, the statement in lines 632-633 may only be correct at the phenomenological or descriptive level but not at the mechanistic or anatomical level. The reason is that those speculations seem to fail to take into account what else the monkeys are doing while making the unilateral arm movements. Unless the monkeys were encased in perfectly form-fitted body casts (which might still not eliminate this problem), all of the movements of each arm will be accompanied by biomechanically-linked bilateral adjustments of the shoulder girdle and axial structures, and even in the ipsilateral arm. These changes would be caused by widely distributed bilateral changes in muscle activity, along with muscle activity that stabilizes different parts of the body against the perturbing effect of the arm movements but does not result in measureable movements. This is inevitable, and would probably be most pronounced in reaching movements of one arm to targets on the other side of the body mid-line. As a result, an undetermined amount of the neural activity recorded during ipsilateral arm movements could result from those biomechanically-linked changes in the contralateral body and arm, rather than being a signal used to control the moving ipsilateral arm itself or some unspecified form of bilateral control. This could also explain an undetermined amount of the deviation of single-neuron arm preference measurements from the theoretically “ideal” extremes of exclusive coupling to only the contra- or ipsilateral arm. Measuring how much the hand of the non-moving arm does or does not move is a completely inadequate control for this (lines 135-136; 159-162). I did not see anything in the manuscript that addressed “any reasonable impacts these small movements have in our neural analyses” (lines 164-165) to my satisfaction. I also don’t find very convincing the argument that arm-specific segregation peaked during the movement phase reduces “any concern that small movements of the non-selected arm had an impact on our results or conclusions” (lines 611-613). I don’t see how their analytic methods can differentiate activity recorded during ipsilateral arm movements that is specifically directed at ipsilateral body targets or alternatively contributes to all the biomechanically-linked adjustments and compensations of the contralateral body during ipsilateral arm movements. If I’m wrong, it would be important to prove that I’m wrong in the manuscript.

3) Perhaps the only experimental approach currently available to deal with this major confound directly while recording neural activity are the heroic techniques that Paul Cheney, Eb Fetz, Roger Lemon and others have used to demonstrate the direct causal effects of spiking activity on peripheral musculature during various tasks. Nevertheless, the authors could do something extremely valuable to start to get a handle on this issue. They could do a thorough study of EMG activity in a number of distal and proximal arm muscles as well as muscles acting on the shoulder girdle, scapula and para-axial structures, while the monkeys made contra- and ipsi-lateral arm movements. It is actually not that difficult to do with percutaneously implanted electrodes or patch electrodes applied to the skin. Paul Cheney’s work gives clear guidance on how to go about it. Ideally, this should be done on the monkeys that participated in this study, if they are still available. They could then process the EMG data in the same manner as the neural data, and look for response patterns in the n-dimensional EMG activity space that are shared with the neurons. Even more informative would be response patterns that are expressed in the neural populations that cannot be extracted from the muscle activity. That would provide a powerful clue into the neural activity that is implicated in arm movement control but cannot be accounted for by biomechanics and muscle activity. This would provide very important experimental data to assess and ultimately justify the validity of such speculations as the statements that “bilaterally distributed networks contribute to computations that do not directly represent the output” (lines 683-684) and “distributed signals could serve to enforce internal dynamics of the overall population” (lines 692-693). There are no solid grounds for those speculations until they can demonstrate that some components of the neural variance patterns are not also seen in the muscle activity at the periphery. If they want to make such speculations, they also need to acknowledge this limitation of interpretation of their neural results. I am not arguing here that the neural data in this manuscript are unpublishable without the EMG data or even that they cannot offer those interpretations until these EMG studies are done. However, I am saying that this issue must be acknowledged and addressed as a factor that limits the interpretation of the present findings. I would also strongly encourage the authors to take on this EMG study, and present a follow-up report of the implications of any similarities and differences in the neural and muscle data. That would be a positively brilliant addition to the field.

4) Bilateral signals (lines 635-654): “…correlations between the tuning for ipsilateral and contralateral arm movements in M1 units tend to be weak or absent….Cisek et al., 2003….Our results are therefore consistent with the functional hypotheses of ipsilateral cortex involving mirror symmetric activation.” I believe that your citation of Cisek et al 2003 is correct for their M1 data. However, if I’m not mistaken, one of their main findings was not just that many more PMd neurons than M1 neurons discharged during movements of both arms, but more specifically that many PMd neurons had similar directional tuning during the instructed delay period of their task independent of the arm used, suggesting an effector-independent representation of the intended motor act in a fixed external spatial reference frame. That would contradict the conclusion in the second quoted sentence about mirror symmetric activity. Of course, Cisek’s findings apply only to their delay period activity in PMd. It is entirely possible that when pooled over the entire population in both PMd and M1, the data extraction techniques used in the present study do reveal a neural response pattern that expresses properties of ipsilateral arm movements that are mirror-symmetric to the predominant contralateral movements. That would be a very cool finding. It’s a pity that the way that they present their data does not allow us to visualize their findings in a way that we can more easily relate to the motor outputs of the tasks, namely the reach movements of each arm to the 6 spatial targets.

5) With that last comment in mind, it would be extremely interesting and useful if the authors could generate “neural space trajectories” of the contralateral-arm and ipsilateral-arm neural data collected during the 6 reaching movements from the dedicated and distributed response components using the 3 most “informative” of their top 4 PCs, similar to what Ames and Churchland (2019) did for their bilateral arm data, and what Shenoy, Churchland, Yu and others have used in a number of previous studies. I know that these displays cannot replace the rigorous quantitative tests that the authors used in this manuscript to establish the statistical validity of their findings. Nevertheless, they complement those analyses by presenting the data in a way that is more graphically intuitive. A pretty picture really is often worth more a thousand words, especially if it is easier to grasp intuitively than a thousand words of descriptions of analyses and statistical test results.

6) Figure 4: This figure provides the summary stats of the neural responses in the task. However, they are based on units of variance, so it is not easy to grasp the meaning of these data distributions in terms of actual neural activity. It is essential to provide concrete examples so that those readers who are still interested in such things, like me, will be able to make that transformation. It would help if the authors would do the following. In Figure 3, add the activity from the Rest phase to the single-neuron plots, and display the time windows for data analysis in each sub-plot. Provide the contra and ipsi modulation strength values for the data in each window for each neuron, and its signed arm preference score for each phase. Then add to Figure 3 the mean M1 and PMd population response histograms for all 3 trial phases for each arm (similar to what they did in Figure 8B), and the resultant population modulation strengths and arm preference scores. Do that separately for the PMd and M1 populations in each monkey. Those additions would be very helpful to understand what the curves in Figure 4 mean, as well as the log scale of Figure 4A. For instance, the Rest phase plot might seem pointless because it will likely be nearly at zero, but will still be useful because the curves in the figure make it look almost the same as the Instruction and Move activity, which is almost certainly misleading. Of course, if you think that the brain is just a gigantic n-dimensional matrix of discharge variances and covariances or orthogonal dimensions in neural activity space, I guess it really doesn’t matter, right? �

7) Figure 4: Wouldn’t the data in Figure 4B be easier to understand if they were plotted as simple frequency histograms rather than cumulative frequency histograms? Does Figure 4C add anything that is significantly different from Figure 4B? I’m not convinced that it does.

8) Lines 213-231 and Figure 4: The main take-home message of this text and the figure is that there is a shift to increased contralateral response correlations from Rest to Instruction and from Instruction to Move. This might be illustrated much more effectively if they plotted the frequency distributions of single-neuron CHANGES in arm preference scores between the phases, rather than the cumulative frequency scores in each phase. They could plot those distributions instead of the plots in Figure 4C, which I do not find particularly useful.

9) Lines 195-196: “….both monkeys entered arm-specific population states during the Rest phase”. Since the activity during the Rest phase is often very low and is quantified in variance units, a change from a mean of 1sp/s to 2sp/s might be highly significant and yield a high arm preference score. However, is that anywhere near as meaningful as a difference between 20sp/s and 40sp/s for ipsilateral and contralateral arm movements respectively during the Instruction or Move phases, even though it might yield a similar arm preference score? Mathematically, the statement is correct, but is that finding functionally important? Provide more justification.

10) Lines 115-117: “…we did observe target-specific information….the subspace containing this information did not segregate signals for the two arms at any stage”. The implications of those statements are unclear. Does this mean that neural activity that expresses differential signals for the 6 targets is confined only to the smaller “distributed” component (e.g., lines 488-493, 560), but is not expressed in the much larger “dedicated” component? The only mentions of target-specific signals that I could find in the manuscript relate to the “distributed” component. Does that mean that the “dedicated” component that accounts for the largest amount of variance discriminates which arm to use, but not what to do with it? If that conclusion is true, that is really surprising and counterintuitive given that at least 1000 studies starting with Evarts has shown the powerful effect of the direction of motor output on M1 and PMd activity. All of those studies would predict that a massive amount of variance should be accounted for by direction and should be coupled to the moving arm. If my conclusion isn’t true, then there is something that I don’t understand about those statements. Which is it? If the former is true, the authors will have to discuss the implications of that surprising finding in depth. If it is not true, then the authors have got to reword those sentences to avoid the confusion.

11) PCA analysis: The entire neural data set comprised sub-sets of simultaneously-recorded neurons with each sub-set recorded in a different daily recording session. It is not clear if they did all their PCA analyses separately only using the neural sub-sets collected simultaneously in each daily recording session, and then somehow pooled across daily sessions. Alternatively, did they do their PCAs by post-hoc pooling of all the neural data collected in different recording sessions into one large data matrix?

12) Equations: Please number all equations in the manuscript sequentially.

13) Expectation: What is the Expectation term “E” in the equation in line 819? The explanation in line 820 does not help. Why change the equation from time “t” in line 801 to “phase” in line 819?

14) 16 data samples (line 815 and elsewhere): If the phase window for data analysis is 300ms and is divided into successive non-overlapping 20ms time bins (lines 796-797), how does that yield 16 data samples per analysis period? My old but still trusty calculator tells me that 300/20=15. Are they actually calculating the instantaneous discharge rates at each 20ms window edge, starting at 0ms, rather than calculating the mean activity within each 20ms window?

15) Figure 4, 5A, 6, 8C,D, S2A: Why did they decide to de-emphasize the data from monkey W by making all of its plots little inserts in the corner of the plots for monkey O? Why not keep the plots the same size for both monkeys? They don’t even plot the corresponding data for monkey W in Figure 5C-E. What do the authors have against poor monkey W? �

16) Dimensionality estimation using cross-validated data reconstruction methods (lines 371-374, and 859-890): Are the authors suggesting by such statements as “this is an essential step to avoid drawing conclusions on noise-dominated dimensions” and “they are prone to over-fitting, which include meaningless components” that the findings of the many PCA-based dimension-reduction studies that did not use those rigorous data processing steps are suspect, if not incorrect?

17) Lines 252-253; 424-428: Explain in simple layman’s terms how neurons with similar discharge rates for movements with both arms can still make different contributions to population-level representations and thus contribute to subspace separation.

18) Figure 7D: the label “Prep” refers to the Instruction phase? If yes, why change the label in that figure?

Reviewer #2: This paper, by Dixon and colleagues titled “Hybrid dedicated and distributed coding in PMD/M1 provides separation and interaction of bilateral arm signals” seeks to understand the nature of bi-hemispheric signals present in PMd and M1. The authors found one each of dedicated and distributed components. They show that the former is primarily responsible for the traditional contralateral arm control at the single unit level, whereas the latter does not discriminate between arms.

I think the paper is well-written, and the analyses are well-thought-out. My only major comments are regarding the interpretation of the results in the context of the recordings that were made.

When recording from different cortical layers, it seems as though there would be guaranteed differences in neural populations based on recording depths. Some discussion on relationship between recording depth and the neural subtypes/ subspaces outlined in this paper would be interesting and useful to the reader. If different populations are primarily located at different recording depths, what does this say about the bi-laterality of these signals? Conversely, what is the interpretation if there is no spatial relationship?

Relatedly, the dimensionality of the neural recordings in this paper is significantly lower than for unilateral recordings made across the surface of M1 and PMd with Utah electrode arrays (Gallego et al. 2020). Does the nature of the recordings in this paper (4 linear probes) lead to to the lower dimensionality? Some discussion of this as compared to arrays that record across the cortical surface would be useful.

Gallego, Juan A., Matthew G. Perich, Raeed H. Chowdhury, Sara A. Solla, and Lee E. Miller. "Long-term stability of cortical population dynamics underlying consistent behavior." Nature neuroscience 23, no. 2 (2020): 260-270.

What analyses, if any, were performed to ensure a single neuron was not recorded across multiple contacts of the linear probes? 100um spacing is rather close, and it is rather plausible the same units are being recorded from multiple contacts (although it is difficult to say for sure, without knowing the size of the contacts on the V-probes).

Minor Comments

Does the larger dimensionality during rest have something to do with the sparse firing during rest? Why is this expected?

What are the red circles in 7B?

**Have all data underlying the figures and results presented in the manuscript been provided?**

Reviewer #1: **No: **I don`t see any link to a repository for the original data sets in the manuscript.

PLOS authors have the option to publish the peer review history of their article (what does this mean?). If published, this will include your full peer review and any attached files.

Reviewer #1: No

Reviewer #2: No

**Have the authors made all data and (if applicable) computational code underlying the findings in their manuscript fully available?**

Reviewer #2: Yes
---

## [Decision Letter · Decision Letter 1]

10 Sep 2021

Dear Dixon,

Sorry for the delay in reaching out.

Thank you for addressing the points raised after the previous submission. Based on the reviews, we are likely to accept this manuscript for publication, providing that you modify the manuscript according to the remaining review recommendations.

Sincerely,

Aldo A Faisal

Associate Editor

PLOS Computational Biology

Daniele Marinazzo

Deputy Editor

PLOS Computational Biology

[LINK]

Reviewer's Responses to Questions

**Comments to the Authors:**

Reviewer #1: The authors have responded positively to the previous reviews, and the article is improved. Nevertheless, I have a few further comments about this revised draft.

I greatly appreciate the new section that the authors added (lines 665-714) about whole-body control and the possibility that a significant portion of the “distributed” activity might be implicated in control of axial and ipsilateral muscles and body parts to deal with the inevitable biomechanical interactions among multiple body parts during unilateral arm movements. I was a bit disappointed that they spend most of the third paragraph trying to discount this as the main origin for that distributed activity. For instance, one of their arguments is that postural effects should be most relevant during movement, which is reasonable, but they recorded that distributed activity even during the delay period. Anticipatory postural adjustments long before the onset of movements in delay tasks is a well-known phenomenon that has been extensively studied in locomotion and postural perturbation studies. I suppose that for people with a strong computational orientation, a “shared distributed computational subspace” might be sexier than boring old biomechanical compensation, but to others like me, neurophysiological mechanisms of whole-body coordination are also quite sexy! :-)

Further on this point and on their later discussion section on bilateral signals (lines 754-798), this discussion almost completely ignores the role of the descending cortico-rubral and cortico-reticular projections that are strongly implicated in postural control and bilateral coordination (except lines 746-748). These neural populations might be contributing to the activity in that distributed neural space.

I also greatly appreciate the new figure of PC-space trajectories (Figure S7) that they added at my request. It is disappointing though that they make only one fleeting reference to it in the text. It could be cited several times because it illustrates very nicely a number of their major points, such as the much greater variance accounted for in the dedicated neural activity compared to the distributed activity, and the differential evolution of the dedicated and distributed activity throughout the trial in PMd versus M1 between the native and opposite hand.

Lines 866-868: This new sentence states that the monkeys started the trials in 1 of 3 different arm positions, either with both hands near the body midline or with one hand near the midline and the positioned more laterally. However, they make almost no further comments about this in this manuscript. Did they pool all neural data across these different starting arm configurations for some of the analyses in this manuscript? For instance, did they generate the velocity traces in Figure 1 from just one starting configuration or pooled across all configurations? I can find one statement that they only used the data from trials with both hands close to the start position for dimensionality reduction (lines 1005-1006). They also stated that they did separate analyses for the data from the three starting configurations for Figures 7, 8 and S7 (lines 1000-1001), but it is not clear whether the results shown in those figures were from only one of the configurations, or pooled across all three. This is very confusing. They state in their response to reviewer comments that they plan to report in more detail the effect of arm starting position on neural activity. I think that they have to make some sort of statement about this potential effect and what impact it might have on the results they are presenting in this manuscript. Does it introduce a confound in the interpretation of the present findings?

Figure 7 and associated text: for some reason, the authors decided to present data in terms of “left arm” and “right arm” instead of ipsilateral and contralateral arm. However, they never explain why they made that change just for that figure, in the Methods (lines 962-964) or anywhere else. Why did they do that?

Lines 202, 231-232: Figure 4 no longer has a part 4C.

Reviewer #2: I believe the authors have appropriately addressed each of my concerns. I appreciate the extra analysis regarding the electrode depth. While there was no consistent relationship, as the authors pointed out this could be the effect of the recording limitations. I believe it would be of great interest to look at this in the future if/when these limitations are removed.

**Have the authors made all data and (if applicable) computational code underlying the findings in their manuscript fully available?**

Reviewer #1: Yes

Reviewer #2: Yes

PLOS authors have the option to publish the peer review history of their article (what does this mean?). If published, this will include your full peer review and any attached files.

Reviewer #1: No

Reviewer #2: No

Figure Files:

Data Requirements:

Reproducibility:

References:

---

## [Decision Letter · Decision Letter 2]

4 Nov 2021

Dear Dixon,

We are pleased to inform you that your manuscript 'Hybrid dedicated and distributed coding in PMd/M1 provides separation and interaction of bilateral arm signals' has been provisionally accepted for publication in PLOS Computational Biology.

Best regards,

Aldo A Faisal

Associate Editor

PLOS Computational Biology

Daniele Marinazzo

Deputy Editor

PLOS Computational Biology

Reviewer's Responses to Questions

**Comments to the Authors:**

Reviewer #1: I have read the re-revised manuscript. The authors have responded adequately to my last round of comments and I have no further comments to add.

Reviewer #2: All of my concerns were addressed with the previous changes.

**Have the authors made all data and (if applicable) computational code underlying the findings in their manuscript fully available?**

Reviewer #1: Yes

Reviewer #2: Yes

PLOS authors have the option to publish the peer review history of their article (what does this mean?). If published, this will include your full peer review and any attached files.

Reviewer #1: No

Reviewer #2: No

---

## [Editor Report · Acceptance letter]

17 Nov 2021

PCOMPBIOL-D-21-00379R2 

Hybrid dedicated and distributed coding in PMd/M1 provides separation and interaction of bilateral arm signals

Dear Dr Dixon,

I am pleased to inform you that your manuscript has been formally accepted for publication in PLOS Computational Biology. Your manuscript is now with our production department and you will be notified of the publication date in due course.

With kind regards,

Livia Horvath
